

# A mobile sensor network to map carbon dioxide emissions in urban environments

Joseph K. Lee[1], Andreas Christen[1], Rick Ketler[1], and Zoran Nesic[1,2]

[1]Department of Geography / Atmospheric Science Program, The University of British Columbia, Vancouver, BC, Canada
[2]Biometeorology Group, Faculty of Land and Food Systems, The University of British Columbia, Vancouver, BC, Canada
*Correspondence to:* A. Christen (andreas.christen@ubc.ca)

**Abstract.** A method for directly measuring carbon dioxide ($CO_2$) emissions using a mobile sensor network in cities at fine spatial resolution was developed and tested. First, a compact, mobile system was built using an infrared gas analyzer combined with open-source hardware to control, georeference and log measurements of $CO_2$ mixing ratios on vehicles (car, bikes). Second, two measurement campaigns, one in summer and one in winter (heating-season) were carried out. Five mobile sensors

were deployed within a $1 \times 12.7\,km$ transect across the City of Vancouver, BC, Canada. The sensors were operated for 3.5 hours on pre-defined routes to map $CO_2$ mixing ratios at street level, which was then averaged to $100 \times 100$ m grids. The grid-averaged $CO_2$ mixing ratios were 417.9 ppm in summer and 442.5 ppm in winter. In both campaigns, mixing ratios were highest in the downtown core and along arterial roads and lowest in parks and well vegetated residential areas. Third, an aerodynamic resistance approach to calculating emissions was used to derive $CO_2$ emissions from the gridded $CO_2$ mixing

ratio measurements in conjunction with mixing ratios and fluxes collected from a 28-m tall eddy-covariance tower located within the study area. These "measured" emissions showed a range of -12 to 226 $kg\,CO_2\,ha^{-1}\,hr^{-1}$ in summer and of -14 to 163 $kg\,CO_2\,ha^{-1}\,hr^{-1}$ in winter, with an average of 35.1 $kg\,CO_2\,ha^{-1}\,hr^{-1}$ (summer) and 25.9 $kg\,CO_2\,ha^{-1}\,hr^{-1}$ (winter). Fourth, an independent emissions inventory was developed for the study area using buildings energy simulations from a previous study and routinely available traffic counts. The emissions inventory for the same area averaged to 22.06

$kg\,CO_2\,ha^{-1}\,hr^{-1}$ (summer) and 28.76 $kg\,CO_2\,ha^{-1}\,hr^{-1}$ (winter) and was used to compare against the measured emissions from the mobile sensor network. The comparison on a grid-by-grid basis showed linearity between $CO_2$ mixing ratios and the emissions inventory ($R^2 = 0.53$ in summer and $R^2 = 0.47$ in winter). 87% (summer) and 94% (winter) of measured grid cells show a difference within $\pm 1$ order, and 49% (summer) and 69% (winter) show an error of less than a factor 2. Although associated with considerable errors at the individual grid cell level, the study demonstrates a promising method of using a

network of mobile sensors and an aerodynamic resistance approach to rapidly map greenhouse gases at high spatial resolution across cities. The method could be improved by longer measurements and a refined calculation of the aerodynamic resistance.

## 1 Introduction

Cities and the cumulative processes of urbanization are key drivers of local and global environmental change (Mills, 2007; Grimmond, 2007). As cities are the centers of increasing population growth and resource consumption, they are also the domi-



nant source of greenhouse gas emissions - in particular carbon dioxide ($CO_2$) - into the atmosphere (Rosenzweig et al., 2010). On the global scale, urban emissions are estimated to contribute up to 20% directly and 80% indirectly to the total anthropogenic $CO_2$ emissions footprint (Satterthwaite, 2008) and are thus responsible for a major proportion of the anthropogenic greenhouse gas emissions that are intensifying positive atmospheric radiative forcing of the troposphere contributing to global climate change (IPCC, 2013).

In cities, the major sources of $CO_2$ are the combustion of fossil fuels for heating, ventilation, and cooling systems (HVAC), transportation, industrial processes, and power generation (Kennedy et al., 2009). Theses fossil fuel emissions are combined with $CO_2$ emitted from biological sources, namely soil, plant and human respiration and in part taken up by photosynthesis of urban vegetation (Christen et al., 2011). Overall, fossil fuel sources dominate in cities, and the uptake by photosynthesis on an annual scale is usually minor, but can be more relevant in summer (Peters and McFadden, 2012; Weissert et al., 2014). The dominance of emissions results in increased concentrations of $CO_2$ in the urban boundary layer (UBL) relative to rural or pristine air (Idso et al., 2001; Grimmond et al., 2002; Vogt et al., 2006). The enrichment of $CO_2$ in the UBL links directly to emissions which are controlled by urban form and function.

With more than 50% of the global population now living in cities (United Nations, 2014), cities are also the place where effective mitigation of climate change, driven by policy, design, and bottom up citizen engagement is possible. IPCC (2014) conclude that the urban scale has the highest potential for agile and sustained implementation of mitigation efforts. Central to the reduction of urban $CO_2$ emissions is the availability of reliable emissions information and inventories and methods of validating city-scale emissions estimates and reduction efforts. While there are a growing number of methods of quantifying emissions in urban areas, there are disconnects between the current spatial and temporal resolution of emissions models, the ever-evolving urban form and function, and block to neighborhood-scale measurements which inform and validate emissions models (Pataki et al., 2009; Kellett et al., 2013). It further remains a challenge to directly measure emissions at fine urban scales and separate emission $CO_2$ measurements in the urban atmosphere into different fossil fuel emissions and biological sources (Christen, 2014).

The overall research goal of this contribution is to develop, apply and test a new methodology to map $CO_2$ emissions in complex urban environments. Our hypothesis is that by data collected from a network of mobile sensors and from an urban eddy-covariance tower can be combined with the aerodynamic resistance approach to calculate and map emissions at fine scales (blocks to neighborhoods) in cities.

Mobile measurement methods have been used in the past for studying the spatial variability of greenhouse gases in cities (Jimenez et al., 2000; Idso et al., 2001; Henninger and Kuttler, 2007; Crawford and Christen, 2014). In general, mobile monitoring methods for greenhouse gases rely on a single, high cost, high precision and accuracy, and bulky sensor systems carried in specialized measurement vehicles (e.g. Brantley et al., 2014). Studies such as those by Tao et al. (2015) and Crawford and Christen (2014) demonstrate mobile systems for monitoring $CO_2$, but most of these systems are still bulky and limited by their cost and installation needs. Therefore most urban studies using mobile approaches utilize sensors that are generally designed for specialized transport vehicles (Bukowiecki et al., 2002; Elen et al., 2013; Crawford and Christen, 2014). While these sys-





tems have the advantage that they can be well equipped with additional components such as calibration tanks or computers, they do not allow for easy deployment and various modes of transport.

There are increasing successes to develop innovative methods for monitoring urban climate and air pollution using pervasive computing and low-cost distributed sensor networks. Top-down data mining approaches using crowd-sourced smart-phone

data have shown the advantage of scalability and data density. For example, Overeem et al. (2011) derived measures of rainfall for the entire Netherlands using the attenuation of a cell phone sender signal to its receiver station. In another example, Overeem et al. (2013) developed methodology to derive fine-scale air temperature measurements using cell phone battery temperatures to examine the urban heat island. Bottom-up approaches using distributed sensor networks have become possible in recent years with the increasing availability of low cost climate and air pollution sensors, open source programmable

microcontrollers, and improvements in networking infrastructure. For example, Meier et al. (2015) used sensor data from a commercial consumer-grade weather station network to examine fine-scale urban heat island effects in the city of Berlin. In another example, Chapman et al. (2015) developed a road sensor network to monitor road surface temperatures to optimally salt roads during the winter months in Birmingham. Given this growing interest in distributed and mobile sensing systems and the advances in low-cost open- and micro technologies, could there be new opportunities for the fine-scale mapping of $CO_2$

emissions in cities? Furthermore, could new methods be developed that are scalable and flexible enough to be integrated into existing infrastructure such as bikes, car-sharing cars, taxis, or even autonomous flying vehicles? Hence, the key considerations for developing new mobile $CO_2$ emission monitoring systems must be around scalability (how many can be built and for what cost?), system extendability (can the system be built upon?), accuracy and precision, temporal resolution, accessibility (e.g open source or proprietary?), and the mobile platform on which the sensor is to be mounted.

The overall research question for this contribution asks whether it is possible to map greenhouse gas emissions, specifically $CO_2$, at a spatial resolution of neighborhoods / blocks across the city with a portable network of mobile sensors that could be routinely implemented on car-sharing platforms, public transit or random vehicles.

In order to address the research question, four major objectives were outlined and developed:

1. Sensor Development: Develop and test a compact, mobile, and multi-modal $CO_2$ sensor for bikes and cars.

2. Measurement Campaign: Deploy the sensors in a targeted measurement campaign.

3. Methodology development: Calculate emissions from measurements of $CO_2$ mixing ratios and aerodynamic resistance (in the following called "measured emissions")

4. Analysis and Evaluation: Compare the measured emissions to fine-scale traffic and building emissions inventories. Can we find agreement between the spatial patterns in the inventories and measured emissions?



## 2   Methods

### 2.1   The DIYSCO$_2$ system

#### 2.1.1   System requirements

A mobile CO$_2$ monitoring system was required to address the project's need for multiple, low cost, yet accurate sensors

capable of measuring mixing ratios and position at high frequency ($\geq$1 Hz) and easily deployable on bikes and various cars
with a compact design. A mobile monitoring system with such specifications is necessary to cover large geographic areas
within limited time scales (hours) at sufficiently fine resolution that are representative of typical urban emission patterns.
Sensor systems with many of these specifications do already exist, but few, if any, were designed to be carried on and easily
interface with various types of mobile platforms; all studies using high accuracy CO$_2$ sensors either have been stationary or

have primarily used specialized vehicles because of the weight, power consumption, and size of the sensors being used and are
highly costly.

#### 2.1.2   System design

Components from the Arduino platform (Arduino CC, Ivrea, Italy), an opensource programmable microcontroller, were cou-
pled with Licor's proprietary Li-820 IRGA (Licor Inc., Lincoln, NB, USA) - a compact (23.23 cm x 15.25 cm x 7.62 cm, 1 kg),

low maintenance (approx. 2 years of continuous use) and high accuracy CO$_2$ ($\pm$ 1 ppm) single-path IRGA built for various
CO$_2$ monitoring applications including agriculture (Li-Cor, 2015) - to prototype a portable CO$_2$ analyzer. The IRGA uses in-
frared light to determine the CO$_2$ mixing ratio within a closed path by detecting the amount of absorption of the light from the
path. With low cost compact components, open code base, and flexible hardware interfacing, the Arduino platform provided a
lightweight and modular prototyping environment capable of communicating digitally with the IRGA, a GPS (Adafruit Ulti-

mate GPS Logger Shield with GPS Module, Manhattan, New York, USA) unit, and digital temperature thermometers (Maxim
Integrated One Wire Digital Temperature Sensor - DS18B20, San Jose, CA, USA). A custom hardware board was developed
to connect all of the components together in a way that: 1. distributes the correct amount of power to each of the hardware
components, 2. allows for hardware and sensor input, and 3. keeps the sensor hardware centralized, organized, and compact.
The portable CO$_2$ analyzer was named the "Do-It-Yourself-Sensor-CO$_2$", or "DIYSCO$_2$" system (Fig. 1a)

The DIYSCO$_2$ system reports CO$_2$ as mixing ratios ($r$) in ppm, geoposition (latitude/longitude, speed, altitude, and satellite
strength), and internal and external air temperature which are logged onto a micro-Secure Digital (SD) card at 1-second
intervals. Air is drawn into the DIYSCO$_2$ system through a 3 m long inlet tube (6.35mm diameter, Dekoron Bendable Tubing,
Mt. Pleasant, Texas, USA) using a small KNF NMP015 Micro-Diaphragm Pump (KNF Neuberger, Inc., Trenton, NJ, USA)
first passing through a mesh filter at the sample inlet head to prevent large particles from entering the DIYSCO$_2$ system (e.g.

insects) and then through a Balston disposable filter unit (DFU) (Parker Hannifin Corporation, Lancaster, NY, USA) at the end
of the 3 m tube. The flow rate is regulated by a Swagelok needle valve at $700 \, \mathrm{cc \, min^{-1}}$ as recommended by Licor to minimize
the effect of internal cell pressure changes on the CO$_2$ measurements. The entire DIYSCO$_2$ system is 35.8 cm x 27.8 cm x





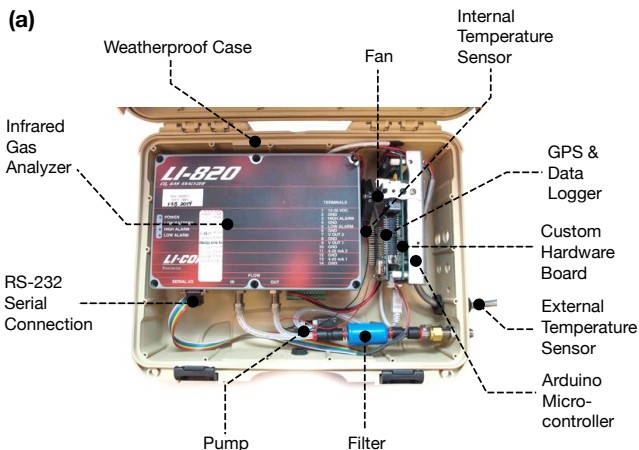

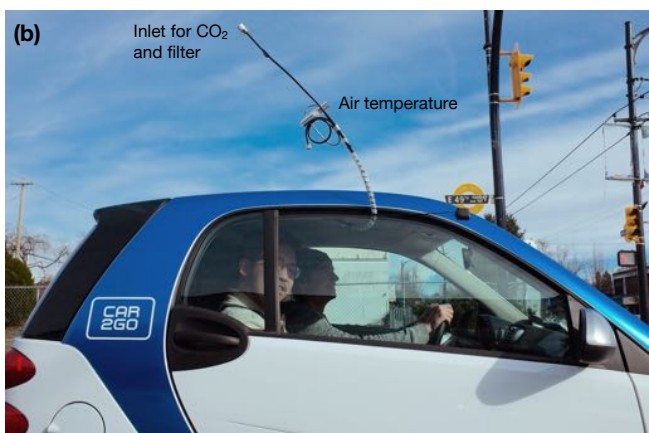

**Figure 1.** (**a**) Photo of the "DIYSCO$_2$" system (case open) with components labelled. During operation, the system is enclosed in the case, while LEDs on the box indicate system state. (**b**) Inlet mounted through the passenger window of a car-sharing vehicle, the "DIYSCO$_2$" sits in the trunk space.



11.8 cm, weighs 2.6 kg and is contained in a weather-proof case (NANUK 910, Plasticase, Terrebonne, CA, USA). The system is powered by a single 9-18V DC/DC input which can be supplied by battery or via car cigarette lighter socket.

### 2.1.3 System testing and installation

Within the range of typical ambient mixing ratios of $CO_2$ between 400 and 550 ppm the $DIYSCO_2$ system showed strong linearity ($R^2$ of 0.9999) and a root mean square error (RMSE) of 0.233 ppm relative to six tanks of reference gases (see Appendix A1). The maximum sensor drift over three hours (the duration of the campaign, see below) under controlled conditions was in the range of -0.31 and +0.51 ppm (see Appendix A2). In the configuration used, the $DIYSCO_2$ had a time lag of 18.2 s between measurement intake and analysis (see Appendix A3).

Appendix A4 discusses errors associated with mounting the inlet at different positions on the car which can lead to a systematic bias. Generally, values on the driver side (centre of road) were higher than the passenger side. In the current work, the sample inlet tube was run out through the passenger side window of the vehicles at the height of 2 m (Fig. 1b). In order to deploy the $DIYSCO_2$ on a bike, the setup requires a 40 ℓ backpack to carry the sensor and a 7 Amp-hour, 12V gel-cell battery and a 1.5 m long rigid mounting tube (6 mm diameter) to mount the inlet tube above the cyclist. The sensor is placed in the backpack with the battery and worn on the back of the cyclist to reduce vibrations to the sensor system.

## 2.2 Measurement campaigns

The systems were tested in two field campaigns. In each of the campaigns, a fleet of five sensors were operated simultaneously on pre-defined routes to evaluate the potential to map emissions and compare them against inventory data.

### 2.2.1 Study area

The study area for testing is a 12.7 km × 1 km quadrangle of diverse urban land uses within the City of Vancouver, BC, Canada (Fig. 2). The study area begins at the northern-most tip of the city (UTM 10,488510 E, 5451513 N) in forested "Stanley Park", and extends to the city's south eastern neigborhood called "Victoria - Fraserview" (UTM 10, 495410E, 5462213N). It includes dominant urban land uses - the downtown core, medium density residential, single detached residential, light industrial development, parks and forest. The study area is encompassing approximately 11.1% of the total area of the City of Vancouver, and was selected, because of the provision of high resolution geospatial data, including LIDAR measurements of urban form used for building emission simulations in previous research (van der Laan, 2011), the availability of detailed traffic counts, and the location of a 30-m tall eddy-covariance tower within the study area.

### 2.2.2 Tower-based measurements

The eddy-covariance tower "Vancouver-Sunset",(ID: Ca-VSu FLUXNET (2016); Crawford and Christen (2015)) is located at the south east corner of the study area (UTM 10, 494273 E, 5452641 E). The eddy-covariance tower was instrumented with a CSAT-3 ultrasonic anemometer-thermomemter (Campbell Scientific Inc., Logan, UT, USA) which provides continuous




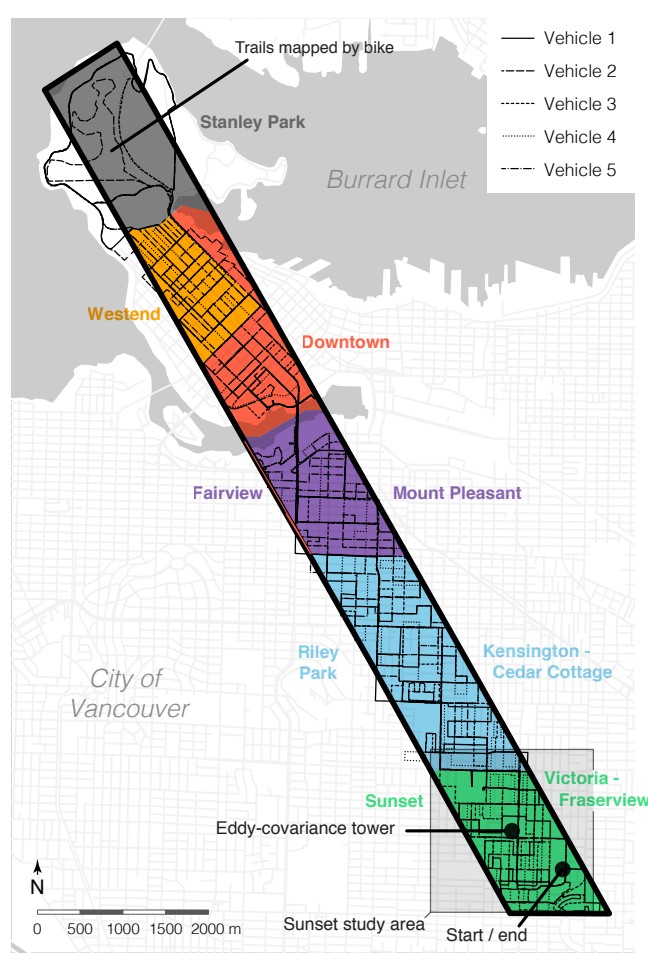

**Figure 2.** Map of the study area, a 12.7 km x 1 km quadrangle (black outline) in the City of Vancouver, BC, Canada. Black lines refer to the paths of each of the five DIYSCO$_2$ systems. The colored areas are the neighborhoods used in further analysis. Shown are also the location of the eddy covariance tower and the start and end point of all paths (where all five systems were cross-checked before and after the campaign). The $1.9 \times 1.9$ km box labelled "Sunset study area" refers to the domain of previous research, including the fine scale emission inventory developed by Christen et al. (2011) and Kellett et al. (2013), and 24 hour measurements of CO$_2$ storage by Crawford and Christen (2014).



measurements of sensible heat flux ($H$), wind direction, wind velocity. Further a shielded HMP 45 thermometer / hygrometer (Vaisala Inc., Vanta, Finland) provided air temperature ($T_{tower}$). A CNR-1 net radiometer (Kipp & Zonen, Delft, The Netherlands) measured all four radiation components including long-wave upwelling radiation ($L_{\uparrow}$). Carbon dioxide molar mixing ratio $r_{tower}$ was measured near tower top (28 m) using a tube that pumps air to a TGA200 closed path analyzer (Campbell Scientific Inc.) and additionally by a Licor-7500 open path IRGA (Licor Inc., Lincoln, NK, USA). The TGA200 is calibrated every 10 minutes against three WMO-traceable tanks of known $CO_2$ mixing ratios to ensure an accuracy of about $< 0.15$ ppm. The Licor-7500 is calibrated twice a year in the lab. Further details of the site location, instrument exposure and data processing are discussed in Crawford and Christen (2015). The availability of this tower made it possible to link mobile measurements with data from above the city and determine aerodynamic resistances for the calculation of emissions (see Section 2.4.1)

### 2.2.3 Mobile measurements

Two field campaigns took place, one on 28 May 2015 (non-heating season, broadleaf vegetation with leaves emerged) and one on 18 March 2016 (heating season, before leaf emergence), both between 10:00 and 13:30 local time. For simplicity, data sets from the two dates will be referred to as "summer" (28 May 2015) and "winter" (18 March 2016). The measurement period was set between 10:00 to 13:30 because this time period was identified to show relatively consistent traffic counts throughout the transect as well as relatively stationary meteorological conditions.

In order to ensure that the study area was comprehensively sampled during the duration of the measurement campaign, transects were predefined for each of the five $DIYSCO_2$ systems (Fig. 2). Taken together, the routes were drawn such that the $DIYSCO_2$ would not only sample some of the same street segments at different times throughout the campaign, but also that a majority of the streets and lanes in the study area would be sampled at least once in the 3.5 hour time period. The predefined routes were evaluated using an overlaid 100 m × 100 m grid, confirming that nearly all of the grid cells would be traversed by at least one system if the routes were successfully completed. Each vehicle was assigned a path to travel approximately 70 km during the study period (achieving an optimal sampling density of about 3.5 $km^2 hr^{-1}$). Each vehicle started and ended at the southeast corner of the transect (UTM 10, 494860 E, 5452010 N, Fig. 2). Furthermore, a bike was used to traverse trails in the forested area of "Stanley Park" to sample in the densely forested ecosystem away from roads.

Five $DIYSCO_2$ systems were installed on vehicles and recorded $CO_2$ mixing ratios $r_{mobile}$, air temperature and GPS location at 1 Hz. Prior to the mobile measurements, all vehicles were parked on the South-Eastern corner of Gordon Park, away from major streets in a school parking lot. The five $DIYSCO_2$ systems were operated for a 15 minute warm up period in their respective vehicles parked next to each other, and then logged for 5 minutes in order to determine their relative offsets before the field campaign; this is called the "in-situ calibration". During the test, all people moved away and 30 m downwind of the vehicles to avoid contamination from human exhaust and all engines were turned off. After the 3.5 hour traverse, all vehicles returned to the starting location, where a second "in-situ calibration" was performed. The data collected in the in-situ calibration was used to determine offsets and drift of the sensors during the campaign. The slope of the senors was determined in the lab the day before each campaign using two reference tanks.





## 2.3 Data analysis

## 2.4 Data post-processing and griding

The 1 Hz-data from all five $DIYSCO_2$ systems were first filtered following the methods in Crawford and Christen (2014). Data were omitted if the GPS speeds were below $5\,\mathrm{km\,h^{-1}}$ (to avoid self contamination by vehicle exhaust when idling). Data were also removed where the IRGA cell temperature and pressure were below $45\,°C$ and $96\,kPa$, to measure within the specifications and calibration of the Li-820.

Vector matrix grids of $50\,m \times 50\,m$, $100\,m \times 100\,m$, $200\,m \times 200\,m$, and $400\,m \times 400\,m$ were mapped onto the study area in a Geographic Information System to spatially aggregate and attribute the $r_{\mathrm{mobile}}$ measured by the $DIYSCO_2$ systems to square grid cells. The separate data analysis for the $50\,m$, $100\,m$, $200\,m$, and $400\,m$ grids provided a way to determine the effects of grid size on emissions estimates. In the results section, the $100\,m$ grid is selected, because the $100\,m$ grid cell size was determined to be significantly large enough to avoid most micro-scale horizontal advection of emissions while also still attributing emissions at a traceable scale to individual arterial roads and features. Appendix C explores the effect of using different grid sizes by comparing the results from the $100\,m$ grid to the $50\,m$, $200\,m$, and $400\,m$ grids.

For each cell, the summary statistics were computed for all valid data points intersecting it. The summary statistics included the mean, median, maximum, minimum, range, skewness, and variance. The gridded data were also classified by neighborhood (Fig. 2) to enable comparisons of $r_{\mathrm{mobile}}$ for areas of different urban form and density. Only grid cells with actual measurements were retained for the analysis. All of the grid cells that did not fall "completely within" the boundaries of the study area were withheld from the analysis.

### 2.4.1 Emission calculation and comparison

Data from the eddy covariance tower is used in conjunction with the gridded averages of $r_{\mathrm{mobile}}$ to calculate emissions for each grid cell based on the aerodynamic resistance approach which posits that the molar flux of $CO_2$ for a given area and time ($\overline{w'c'}$ in $\mu\mathrm{mol\,m^{-2}\,s^{-1}}$) is equal to the difference of the molar concentration $c$ (in $\mu\mathrm{mol\,m^{-3}}$) at the height above the RSL ($c_{\mathrm{tower}}$) and screen level at $2\,m$ height ($c_{\mathrm{mobile}}$) divided by the aerodynamic resistance of $CO_2$ (in $\mathrm{s\,m^{-1}}$):

$$\overline{w'c'} = -\frac{c_{\mathrm{tower}} - c_{\mathrm{mobile}}}{r_{aC}} \tag{1}$$

While both, $c_{\mathrm{tower}}$ and $c_{\mathrm{mobile}}$ are available through the measurement of $r$ and density (considering pressure and air temperature), the challenge is that $r_{aC}$ cannot be directly and easily measured due to the spatial heterogeneity of $\overline{w'c'}$ and $c_{\mathrm{mobile}}$. Hence, to make the approach more robust, it uses the availability of sensible heat flux $H$ ($\mathrm{W\,m^{-2}}$), air temperature at $24\,m$ height ($T_a$) and surface brightness temperatures ($T_0$). This is possible because a city is a relatively homogeneous source of sensible heat and temperatures are more uniform than $CO_2$ fluxes and mixing ratios. From the tower measurements of air





temperature ($T_a$) and surface brightness temperature we then calculate the aerodynamic resistance of sensible heat $r_{aH}$ (Kanda et al., 2007). $r_{aH}$ is the integral resistance from the surface (ground, roofs) to the top of the tower.

$$r_{aH} = \rho c_p \frac{T_{\text{tower}} - T_0}{H} \tag{2}$$

where $T_{\text{tower}}$ is the air temperature (K) at the height of the tower (24 m), $T_0$ is the surface brightness temperature (in K,

calculated as $T_0 = (L_\downarrow/\sigma)^{0.25}$) from the long-wave radiometer, where $\sigma = 5.6 \times 10^{-8}\,\text{W}\,\text{m}^{-2}\,\text{K}^{-4}$ is the Stefan-Boltzmann constant), and $H$ is the sensible heat flux ($\text{W}\,\text{m}^{-2}$) measured by eddy covariance.

In a next step we assume Reynolds analogy, which assumes equivalency of scalar transfer, i.e. that the aerodynamic resistance of sensible heat is equal to the aerodynamic resistance of carbon dioxide ($r_{aC}$) and rewrite Eq. 1.

In order to convert the molar flux $\overline{w'c'}$ (in $\mu\text{mol}\,\text{m}^{-2}\,\text{s}^{-1}$) to a mass flux $F_c$ consistent with inventories (in $\text{kg}\,\text{CO}_2\,\text{ha}^{-1}\,\text{hr}^{-1}$),

we rewrite:

$$F_c = -M_c\, b_a\, b_t\, b_o\, b_m\, \frac{c_{\text{tower}} - c_{\text{mobile}}}{r_{aH}} \tag{3}$$

where $M_c$ is the molar mass of $CO_2$ (44.01 $\text{g}\,\text{mol}^{-1}$), $b_a$ is a factor for converting $\text{m}^{-2}$ to $\text{ha}^{-1}$ (i.e. $b_a = 10^4\,\text{m}^2\,\text{ha}^{-1}$), $b_t$ is a factor for converting $\text{s}^{-1}$ to $\text{hr}^{-1}$ (i.e. $b_t = 3600\,\text{s}\,\text{hr}^{-1}$), $b_o$ is the factor for converting $\mu\text{mol}$ to mol (i.e. $b_m = 10^{-6}\,\mu\text{mol}\,\mu\text{mol}^{-1}$) and $b_m$ is the factor for converting g to kg (i.e. $b_m = 10^{-3}\,\text{kg}\,\text{g}^{-1}$).

Equation 3 was applied to each grid cell in the two measurement campaigns, where $c_{\text{mobile}}$ varied for each grid cell and each time, while $r_{aH}$ and $c_{\text{tower}}$ varied only over time. The calculated emissions $F_c$ are then compared to independent gridded building and traffic emissions estimates to test the feasibility and accuracy of the method (the derivation of the independent emissions inventories is documented in Appendix B.

In summary, this procedure to calculate emissions from mobile and tower measurements is only valid under the following

key assumptions:

1. $CO_2$ concentrations in the well mixed UBL (the tower location) at daytime will not change dramatically over a short time period or space (e.g. over 30 min time periods are long enough where urban fluxes are well represented) given the same meteorological conditions and are therefore in an equilibrium. In other words, the measurements of $c_{\text{tower}}$ are representative of the UBL above each grid cell at any time.

2. The flux at the height of the tower is directly related to the flux at the surface, hence concentration changes over time in the layer between surface and tower are negligible at day (i.e. no storage flux). This assumption is supported for the daytime by independent measurements documented in Crawford and Christen (2014) for the current study area and in Helfter et al. (2011) for a higher-density area in Central London, UK. Bjorkegren et al. (2015) and Crawford and Christen (2014) and also conclude that this assumption is severely violated at night and in the early to mid morning, so

the proposed approach would only work midday or afternoon.





**Table 1.** Summary of weather conditions during the two campaigns (from 09:00 to 13:00 PST) measured on top of the urban climate tower "Vancouver-Sunset" (Ca-VSu) located within the study transect

|  | Summer | Winter |
|---|---|---|
|  | 28 May 2015 | 18 March 2016 |
| Surface temperature | 31.0 °C | 15.2 °C |
| Relative humidity (26.0 m) | 71.5% | 36.2% |
| Solar irradiance (26.2 m) | 817 W m$^{-2}$ | 475 W m$^{-2}$ |
| Net radiation (26.2 m) | 680 W m$^{-2}$ | 323 W m$^{-2}$ |
| Sensible heat flux (28.8 m) | 390 W m$^{-2}$ | 120 W m$^{-2}$ |
| Wind speed (28.8 m) | 2.6 m s$^{-1}$ | 1.9 m s$^{-1}$ |
| Wind direction (28.8 m) | 237° | 70° |
| $CO_2$ mixing ratio (28.8 m) | 396.6 ppm | 420.2 ppm |

3. Reynolds analogy applies to $r_{aC} = r_{aH}$ and $r_{aH}$ and therefore $r_{aC}$ is constant across all the urban densities/local climate zones (LCZs) in the study area/city. Despite the fact that there are varying urban densities throughout a city, the idea is that the resistance will not change significantly.

4. Lateral Advection of $CO_2$ between the surface and the height of the tower in-between grid-cells are negligible, or at least add random (unbiased) noise.

## 3 Results

### 3.1 Field campaign

Weather conditions on both dates were cloudless, convective and steady. Table 1 summarizes the weather and environmental conditions for the two campaigns.

### 3.1.1 Raw data points

A total of 41,027 1 Hz-measurements were available in summer and 42,786 measurements in winter from the 5 DIYSCO$_2$ systems during a 3.5 hour window after filtering. Fig 3 shows the frequency distribution of the filtered 1 Hz $r_{mobile}$ measured by all five DIYSCO$_2$ systems alongside the mixing ratio on the tower ($r_{tower}$).

In summer, the measured 1 Hz $r_{mobile}$ were ranging from 380.2 ppm to 918.1 ppm with a median and average $r$ of 408.5 ppm and 419.5 ppm (std. dev. 32.35 ppm) respectively for the entire dataset. The lowest $r_{mobile}$ (<400 ppm) were measured in the forest at "Stanley Park", in select well vegetated residential streets, and in a large cemetery. The highest values (>800 ppm) were measured in "Downtown" and along the major transport corridors such as "Knight St." (Fig. 4) and "West Georgia St."



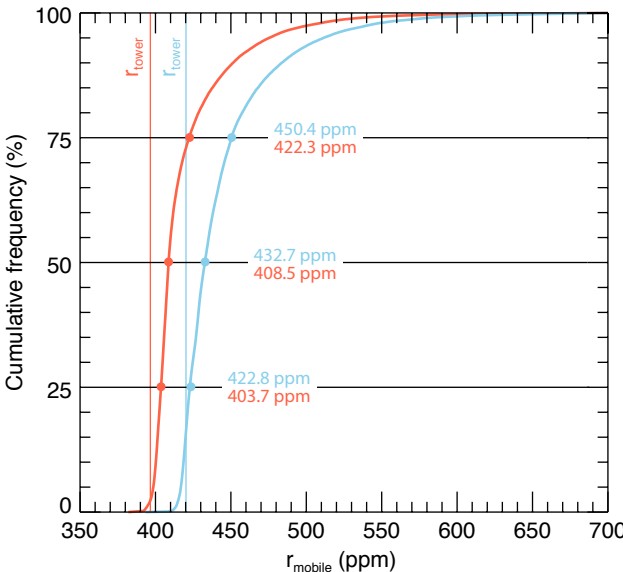

**Figure 3.** Cumulative frequency distribution for raw 1-second $r$ measured by all five mobile systems in the summer (red) and winter (blue) campaign. The thin vertical lines correspond to the average $r$ on top of the tower during the period of the campaign. The colored numbers on the horizontal lines refer to the 25%, 50% and 75% percentiles for summer (red) and winter (blue).

(Highway 99). In winter, overall $r$ were higher for both tower and mobile system. In winter, the measured 1 Hz $r_{\text{mobile}}$ were ranging from 401.4 ppm to 918.5 ppm with a median and average $r_{\text{mobile}}$ of 432.7 ppm and 443.9 ppm (std. dev. 34.77 ppm).

2% and 16% of the measured $r_{\text{mobile}}$ were lower than the tower ($r_{\text{tower}}$) during the summer and winter campaign, respectively. 3% and 7% were higher than 500 ppm in summer and winter, respectively.

## 3.1.2 Grid sample counts

For the 100 m × 100 m grid cells that could be traversed, in summer 91.31% of the grid cells contained more than 10 samples per grid cell, 69.24% of cells contained more than 20 samples, and 28.32% of cell contained more than 50 samples. For the winter campaign, 90.85% of the grid cells contained more than 10 samples, 72.64% contained more than 20 samples, and 27.36% contained more than 50 samples. Grid cells with less than 10 samples were removed from further analysis. Generally, grid cells along major roads tended to have more sample counts because they were traversed at different times, often by different vehicles.

## 3.1.3 Grid averaged statistics

Of the 1332 grid cells that could be traversed by a car or bike, the case study covered 1024 in summer and 1037 in winter, of which 821 and 856 were further used (based on the condition of more than 10 samples). The maps of gridded $r_{\text{mobile}}$ for the summer and winter campaign are shown in Fig. 5. Table 2 summarizes the measured mixing ratios separated by neighborhood.







**Figure 4.** 3D-visualization of all raw $r_{\text{mobile}}$ measurements from all systems (summer campaign) in the "Sunset / Victoria-Fraserview" neighborhood. The visualization is illustrating the high density of measurements taken along streets, laneways, and in parks. The linear area with many higher mixing ratios is the busy 6-lane "Knight St". with $\approx 50,000$ vehicles per day. Image visualized in Google Earth.



**Table 2.** Grid-averaged mixing ratios ($r_{mobile}$), standard deviation of all grid cell means in the neighborhood, and fraction of cells with $r_{mobile} < r_{tower}$ per neighborhood

| Neighborhood | LCZ[a] | Mean mixing ratio $r_{mobile}$ (ppm) | | Std. dev. of $r_{mobile}$ (ppm) | | Fraction of cells with $r_{mobile} < r_{tower}$ | | Number of grid cells | |
|---|---|---|---|---|---|---|---|---|---|
| | | Summer | Winter | Summer | Winter | Summer | Winter | Summer | Winter |
| Stanley Park | A | 413.7 | 435.6 | 19.1 | 24.3 | 4% | 28% | $N = 78$ | $N = 86$ |
| West End | 1 | 416.1 | 442.7 | 15.1 | 15.9 | 1% | 4% | $N = 102$ | $N = 111$ |
| Downtown | 1 | 437.8 | 474.9 | 19.2 | 26.5 | 0% | 0% | $N = 117$ | $N = 115$ |
| Fairview / Mount Pleasant | 6 & 8 | 421.2 | 446.2 | 19.0 | 17.6 | 0% | 0% | $N = 136$ | $N = 144$ |
| Kensington-C. C. / Riley Park | 6 | 411.0 | 432.3 | 13.5 | 15.1 | 1% | 11% | $N = 225$ | $N = 245$ |
| Sunset / Victoria-Fraserview | 6 | 413.3 | 434.7 | 14.2 | 16.0 | 0% | 8% | $N = 163$ | $N = 155$ |

[a] "LCZ" refers to the dominant local climate zones in the neigborhood according to Stewart and Oke (2012).

In summer, the grid averaged $r_{mobile}$ of all valid gird cells in the entire transect ranged between 393.1 ppm and 518.0 ppm, averaged 417.9 ppm, and had a median of 410.0 ppm. In winter, the grid averaged $r_{mobile}$ ranged between 408.4 ppm and 560.5 ppm, averaged 442.5 ppm. 3% of all grid cells in summer, and 8% in winter were showing a $r_{mobile}$ that was lower than $r_{tower}$, the majority of those cases were located in the forested "Stanley Park" in both campaigns (Tab. 2). Selected cells in the

residential parts of "Riley Park / Kensington - Cedar Cottage" neighborhood were also showing a $r_{mobile}$ that was lower than $r_{tower}$.

Both campaigns showed considerable variation of $r_{mobile}$ between grid cells in the same neighborhoods. Overall, the grid cells covering major arterial roads and downtown core showed the highest maximum, minimum, median and mean $r_{mobile}$. Conversely, the grid cells covering residential streets and forested trails exhibited the lowest $r_{mobile}$ for the same statistics.

Of all neighborhoods, "Kensington-Cedar Cottage / Riley Park" exhibited the lowest, and "Downtown" the highest average $r_{mobile}$ in both campaigns (Tab. 2).

Similarly, standard deviations within each 100 m grid cell (not shown) are highest along the major arterial roads and in "Downtown". In contrast, the residential areas have lower standard deviations within grid cells indicating less variability in $r_{mobile}$ for less busy roads. The trends are similar in the winter campaign except that there is overall higher standard deviation

in the residential areas compared to the summer campaign. Over 65.98% of the cells in summer and 66.80% in winter had a positive skewness which means there are intra-grid peaks in measured $CO_2$ mixing ratios.

### 3.1.4    Measured emissions

The aerodynamic resistance $r_{aH}$ for each measurement campaign was calculated by averaging $H$, averaging $T_0$, and averaging $T_{tower}$ over the 3.5 hours of the field campaign. The resulting $r_{aH}$ was $34.14\,\mathrm{s\,m^{-1}}$ in Summer and $56.12\,\mathrm{s\,m^{-1}}$ in winter.

The measured $CO_2$ emissions calculated using Eq. 1 showed a range of -12.0 kg $CO_2$ ha$^{-1}$ hr$^{-1}$ (net uptake) to 225.6 kg $CO_2$ ha$^{-1}$ hr$^{-1}$ in the summer campaign and -13.7 to 162.4 kg $CO_2$ ha$^{-1}$ hr$^{-1}$ in winter. The median and average





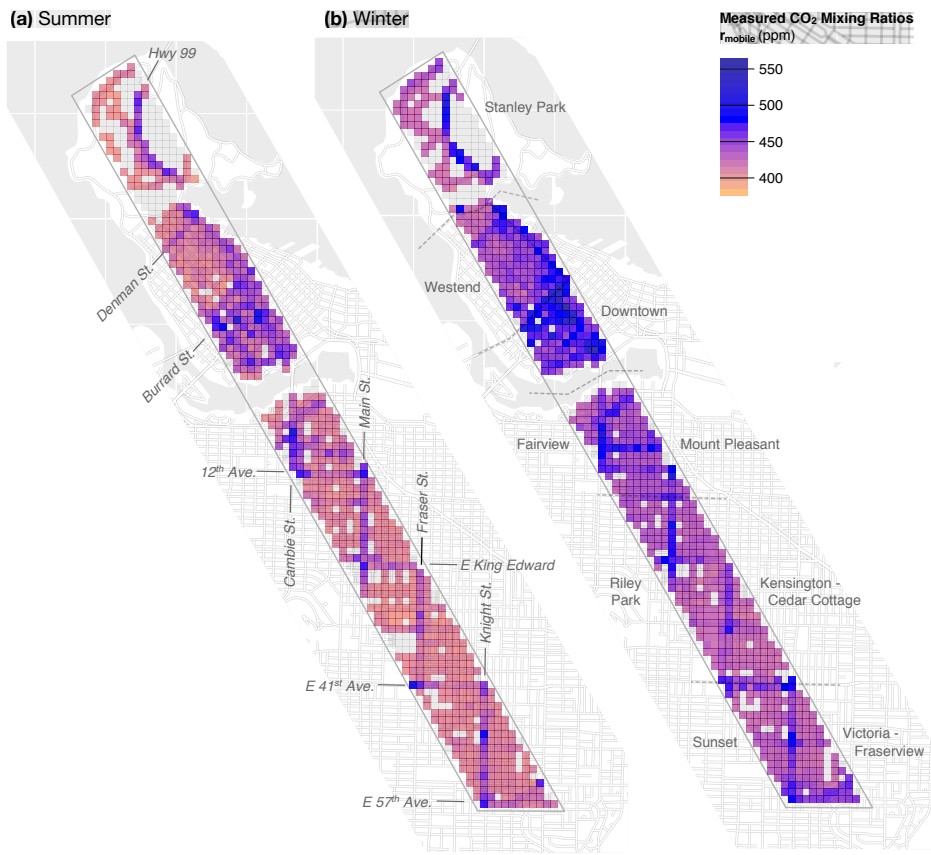

**Figure 5.** Map of grid-averaged $CO_2$ mixing ratios ($r_{mobile}$) for (a) summer and (b) winter campaign using the same scale. The grid size is $100 \times 100$ m.

emissions were respectively 20.1 and 35.0 $\mathrm{kg\,CO_2\,ha^{-1}\,hr^{-1}}$ for the summer campaign and 17.1 and 25.6 $\mathrm{kg\,CO_2\,ha^{-1}\,hr^{-1}}$ for the winter campaign. Highest emissions in general were located in "Downtown" and along the major transport corridors and intersections (Fig. 6, Tab. 3).

## 3.2 Comparison to emissions inventory

### 3.2.1 Characteristics of emissions inventories

The gridded traffic emissions inventory at $100\,\mathrm{m} \times 100\,\mathrm{m}$ resolution (see Appendix B1 and Fig.7a) showed median and mean emissions respectively of 2.37 and 12.50 $\mathrm{kg\,CO_2\,ha^{-1}\,hr^{-1}}$ for the summer campaign and 2.17 and 12.19 $\mathrm{kg\,CO_2\,ha^{-1}\,hr^{-1}}$ for the winter campaign. As expected, the major roads and the areas with the densest road network (e.g. "Downtown") exhibited the highest emissions, all of which were greater than 18 $\mathrm{kg\,CO_2\,ha^{-1}\,hr^{-1}}$. The greatest traffic emissions in a single grid cell was 123.60 $\mathrm{kg\,CO_2\,ha^{-1}\,hr^{-1}}$.





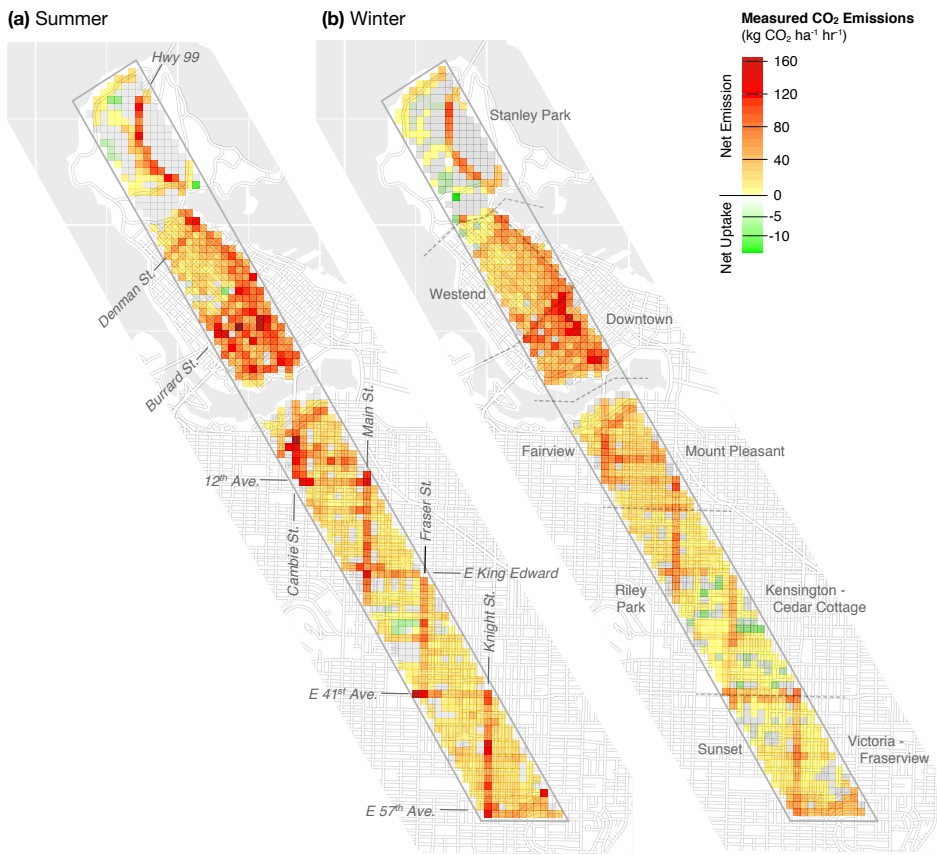

**Figure 6.** Measured emissions (calculated from mixing ratios using the aerodynamic resistance approach in Eq. 1) for (a) summer and (b) winter campaign at a resolution of $100 \times 100$ m.

The building emissions inventory (see Appendix B2), is shown in Fig.7b. In summer, the data for the 100 m grid showed a median and mean of 6.69 and 10.19 $\mathrm{kg\,CO_2\,ha^{-1}\,hr^{-1}}$, respectively. In winter, the data for the 100 m grid showed a higher median and a higher mean of 13.08 and 20.44 $\mathrm{kg\,CO_2\,ha^{-1}\,hr^{-1}}$, respectively. The maximum rate of building emissions was located in "Downtown". The building emissions inventory only covers a subset of the transect area (Fig. 7b). Data for part of

5    "West End" and for "Stanley Park" are not available.

The total emissions inventory is the sum of the building and traffic emissions estimates (Fig.7c). For the summer campaign, the median and mean of the total emissions estimates were 10.15 and 22.06 $\mathrm{kg\,CO_2\,ha^{-1}\,hr^{-1}}$, respectively. Overall, for the area with both inventories available, 59% of the emissions were estimated from traffic and 41% from buildings. For the winter campaign, the total emissions estimates were 15.87 and 28.76 $\mathrm{kg\,CO_2\,ha^{-1}\,hr^{-1}}$, respectively, and 41% of the emissions were

10    estimated from traffic and 59% from buildings. The fraction of traffic emissions is higher in the detached residential areas (LCZ 6 and 8) and lower in "Downtown" (Tab. 3).



**Table 3.** Comparison of measured emissions, with inventory emissions, separated by neighborhood based on a $100 \times 100$ m grid.

| Neighborhood | Measured emissions (kg $CO_2$ ha$^{-1}$ hr$^{-1}$) | Emission inventory (kg $CO_2$ ha$^{-1}$ hr$^{-1}$ | Relative error (RE) | Mean absolute error (MAE) (kg $CO_2$ ha$^{-1}$ hr$^{-1}$) | Fraction of traffic | Grid cells |
|---|---|---|---|---|---|---|
| **Summer** | | | | | | |
| West End | 47.6 | 30.4 | +56% | 29.3 | 34% | $N = 21$ |
| Downtown | 75.1 | 63.3 | +19% | 28.9 | 54% | $N = 90$ |
| Fairview / Mount Pleasant | 41.4 | 27.4 | +51% | 19.7 | 70% | $N = 136$ |
| Kensington-C. C. / Riley Park | 21.9 | 14.5 | +51% | 10.9 | 60% | $N = 225$ |
| Sunset / Victoria-Fraserview | 26.5 | 13.3 | +99% | 15.3 | 73% | $N = 162$ |
| **Winter** | | | | | | |
| West End | 30.1 | 43.4 | -31% | 24.8 | 22% | $N = 24$ |
| Downtown | 65.3 | 92.1 | -29% | 41.6 | 35% | $N = 92$ |
| Fairview / Mount Pleasant | 30.3 | 34.7 | -13% | 14.6 | 52% | $N = 142$ |
| Kensington-C. C. / Riley Park | 14.0 | 19.4 | -28% | 10.1 | 40% | $N = 244$ |
| Sunset / Victoria-Fraserview | 16.8 | 17.1 | -2% | 12.4 | 56% | $N = 155$ |

### 3.2.2 Mixing ratios vs. emissions inventory

First, measured $r_{\mathrm{mobile}}$ were compared to the emissions estimates to identify if there is a direct relationship between measured mixing ratios and hourly emissions estimates from the emissions inventory. It is observed that as emissions in the inventory increase, the range of the measured $r_{\mathrm{mobile}}$ becomes greater. The relationship between measured $r_{\mathrm{mobile}}$ and traffic shows

5    generally a linear correlation (Fig. 8a and b). Further, measured $r_{\mathrm{mobile}}$ and building emissions are also positively correlated, but with more scatter (Fig. 8c and d). Best agreement is achived when comparing $r_{\mathrm{mobile}}$ to the total (i.e. traffic + building) emissions (Fig. 8e and f). The linear equations given in Fig. 8e show $R^2 = 0.53$ in summer and $R^2 = 0.47$ in winter.

### 3.2.3 Measured emissions vs. emissions inventory

Figure 9a and b show the measured emissions as a function of the traffic emissions inventory. The data show that 86.71% of

10    the measured emissions are within a factor of $\pm$ 10 of the traffic emissions estimates for 100 m grids for the summer campaign (grey shaded area in 9). For the winter campaign, 93.74% of the measured emissions are within a factor of $\pm$ 10 of the traffic emissions estimates for 100 m grids. In particular in areas with lower traffic emissions and where the urban density is lower (e.g. "Sunset / Victoria-Fraserview") the measurements are higher than the emission inventory (note that building emissions are not considered in Fig. 9a and b). The measured emissions and the traffic emissions inventory were found to be correlated

15    positively by 77.87% for the 100 m grid in the summer campaign and 71.75% in the winter campaign.





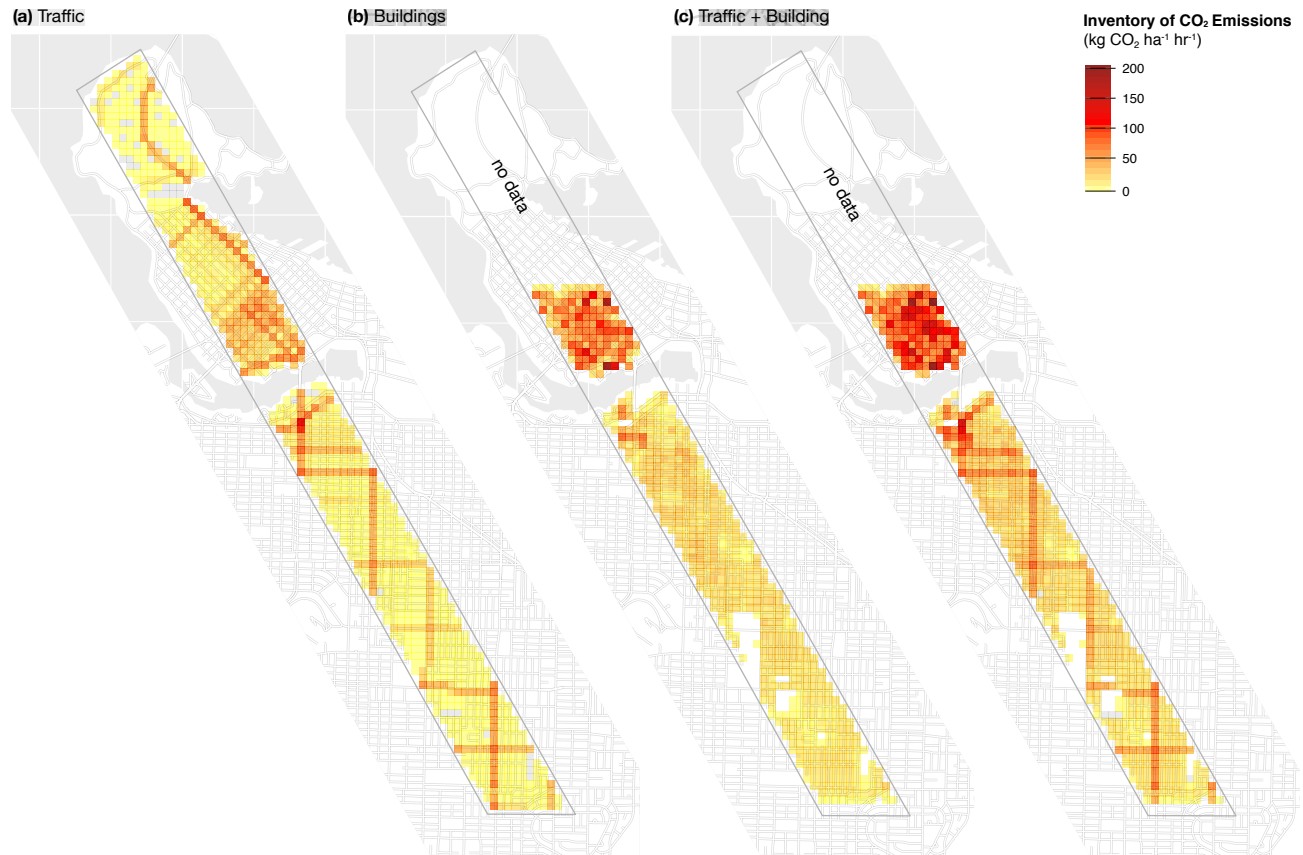

**Figure 7.** Emission inventory for (a) traffic emissions, (b) local building sector emissions, and (c) total (traffic + buildings) emissions for the time of the winter campaign. The equivalent emission inventory for the summer date (not shown) does not look significantly different, but has overall lower building emissions. Note that the building inventory, available from a previous study, did not extend into the Northern part of the transect (label "no data") due to lack of high-resolution LIDAR data in this part of the city.

In Fig. 9c and d measured emissions and the building emissions inventories are compared for each grid-cell. Building emissions are clustered by neighborhood with the lowest urban density (LCZ 6) of "Sunset / Victoria-Fraserview" exhibiting the lowest emissions and "Downtown" with the highest urban density (LCZ 1) exhibiting the highest building emissions. Across all neighborhoods, the measured emissions are higher than the building emissions only (note that traffic emissions are not considered in Fig. 9c and d). The measured emissions and the building emissions estimates were found to be correlated positively by 35.91% for the 100 m grid in the summer campaign and 32.42% in the winter campaign.

Last, Fig. 9c shows the measured emissions as a function of the total emissions (building + traffic) inventory. For the summer campaign the data show that 86.71% of the measured emissions are within a factor of $\pm$ 10 of the total emissions estimates for 100 m grid. The measured emissions and the total emissions inventory were found to be correlated positively by 77.87%



**Figure 8.** (left column: a,c,e) Comparison of inventory (traffic only, building emissions only, and total emissions) against grid-averaged mixing ratios ($r_{mobile}$) where each dot is a $100 \times 100$ m grid cell. Note that the x-axis is logarithmic. The curves in (e) are linear fits (Right column: b,d,f). Comparison of inventory (traffic only, building emissions only, and total emissions) to the difference between grid-averaged mixing ratio $r_{mobile}$ and the mixing ratio measured at the tower.







**Figure 9.** Comparison of inventory emissions and measured emissions on a grid-by-grid basis plotted with double logarithmic axes. The black line is the 1:1 curve and the grey area shows data within one order of magnitude of each other. Grid cells with less than $0.1\,\mathrm{kg\,CO_2\,ha^{-1}\,hr^{-1}}$ in the emission inventory and/or measured emissions are not shown. $n$ refers to the number of grid cells included in the comparison.





for the 100 m grid. For the winter campaign, the data show that 92.58% of the measured emissions are within a factor of $\pm$ 10 of the total emissions estimates for 100 m grid. The measured emissions and the total emissions inventory were found to be correlated positively by 71.75% for the 100 m grid.

Across all valid grid cells in the study area, the measured emissions in summer averaged to 35.11 $\mathrm{kg\,CO_2\,ha^{-1}\,hr^{-1}}$ as compared to 22.06 $\mathrm{kg\,CO_2\,ha^{-1}\,hr^{-1}}$ of the emissions inventory. In winter, the measured emissions in averaged to 25.92 $\mathrm{kg\,CO_2\,ha^{-1}\,hr^{-1}}$ as compared to 28.76 $\mathrm{kg\,CO_2\,ha^{-1}\,hr^{-1}}$ of the emissions inventory.

In summer, 73% of the grid cells show measured emissions that are greater than the corresponding grid cells of the total emissions inventory. For the winter campaign, only 35% of the measured emissions are greater than the total emissions inventory. For both the summer and winter campaigns, emission measurements are higher than inventory in grid cells along major arterial roads whereas the measurements are lower than the inventory in residential areas and in "Downtown".

The mean absolute error (MAE) for all grid cells in the entire transect between measured and modelled total emissions is 17.1 in summer and 16.6 in winter. The median absolute error for the entire transect is 9.6 in summer and 9.9 in winter. Table 3 lists the MAE by neighborhood. The MAE is about a factor 2 larger in "Downtown" and "West End" compared to the residential and industrial neighborhoods.

The relative error (RE) is defined as the difference between a grid cell's measured emission and the same cell's emissions inventory divided by the cell's emissions inventory. The data for the 100 m grid show that 62% of the grid cells in summer and 81% in winter have a RE within a factor of $\pm 1$. As expected, locations with higher relative errors were locations in which the building and traffic emissions inventories estimated almost zero but measured emissions were higher. When excluding grid cells with emissions $< 10\,\mathrm{kg\,CO_2\,ha^{-1}\,hr^{-1}}$) in the inventory, 80% of the grid cells in summer, and 91% in winter have a RE with a magnitude of less than $\pm 1$.

## 4 Discussion

### 4.1 Assessment of the measurement methodology

Overall, the developed approach lead to realistic and consistent results. The spatial patterns of measured emissions are plausible and match generally the fine-scale inventories of traffic and buildings although at the scale of an individual grid cell, large errors up to an order of magnitude are observed. The study was also able to replicate in the winter campaign the spatial patterns and the magnitude found in summer. The results demonstrate the potential to apply an aerodynamic resistance approach to measuring emissions using a network of mobile sensors and data from an urban climate tower.

Building and traffic emissions are both good predictors of $r_{\mathrm{mobile}}$ measured in a city at ground level. This implies that values of $r_{\mathrm{mobile}}$, from microscale to neighborhood scales, are related the $CO_2$ emissions being generated at those scales (and presumably this also holds for primary, less reactive air pollutants). This suggests that it is possible to link $r$ to emissions across a complex landscape under specific, stationary atmospheric conditions. Nevertheless, several challenges remain.

Overall, the building emissions were less clearly correlated with the spatial variability in $r$ than traffic emissions which were a better predictor. Building emissions of $CO_2$ (natural gas burning) are most likely injected into the atmosphere at roof





level (chimneys), where higher winds blend them in the process of downward mixing into streets and laneways where mobile sensors were operated. As a result of this blending, the signal of $r$ might show less spatial variability if emissions originate from buildings (far from sensor) compared to situations near ground-level emissions (car exhaust on arterial roads). Measured emissions generally tend to underestimate the inventory in "Downtown" where there are a high density of tall buildings that

vent their emissions usually at higher storeys, likely decoupled from the grid cells at ground. Consequently, the observed peaks in $r$ are more likely to be a result of traffic emissions alone.

Despite these differences data aligned relatively well with an independent previous study by Christen et al. (2011) that measured and modelled emissions for the $1.9 \times 1.9$ km study area surrounding the "Vancouver-Sunset" tower (see Fig. 2). In the study area, the annual total $CO_2$ emissions were modelled to be $26.87$ kg $CO_2$ ha$^{-1}$ hr$^{-1}$ and validated using direct

eddy-covariance measurements of $CO_2$, which were on average $25.96$ kg $CO_2$ ha$^{-1}$ hr$^{-1}$ over the year. The current study estimates emissions for the "Sunset / Victoria-Fraserview" neighborhood (that overlaps with the area in (Christen et al., 2011)) as $21.65$ kg $CO_2$ ha$^{-1}$ hr$^{-1}$ (average of summer and winter campaigns). Note that the time scales of the two studies disagree. Christen et al. (2011) report annual and monthly emissions, while the current study is restricted to weekdays between 10:00 and 13:30.

In selected areas negative net ecosystem exchange (NEE) were detected, such as in the forest at "Stanley Park", in some highly vegetated urban residential areas and the lawn area of a cemetery. This is plausible, because most grid cells have likely some uptake by photosynthesis of urban vegetation, but in many cells the emissions from combustion and respiration combined are greater than photosynthesis. In comparing our lowest measured emissions from "Stanley Park" (-12 kg $CO_2$ ha$^{-1}$ hr$^{-1}$) to a study by Humphreys et al. (2006) who measured NEE for a forest with similar stand composition (Douglas Fir forest on

Vancouver Island, 200 km to the W) in April and June in the same latitude. We find that our measured emissions were within a factor of 2 of those observed in a typical forest at the same time of day and year.

## 4.2   Possible refinements and errors

Ultimately, the comparison of measured emissions and the emissions inventories showed where there might be close alignment or divergences between the datasets and suggests promising new research opportunities for improving the proposed methodol-

ogy and/or emissions inventories.

### 4.2.1   Aerodynamic resistance

In terms of methodology, $r_{aH}$ is calculated using $T_{\text{tower}}$ and $T_0$ at a single location, likely not representative for the entire city. There is evidence of varying aerodynamic resistances across the study area. For example in the narrow street canyons of "Downtown" and in forested "Stanley Park", it is likely that the aerodynamic resistance is higher, because of the sheltered

nature of the deep canyons and forest canopy, respectively. Generally, measured emissions could possibly be overestimated in streets with a denser tree canopy regardless if the canopy is vegetation or buildings. An area with a dense tree canopy may actually reduce mixing (Jin et al., 2014) and as a result, the measured $r_{\text{mobile}}$ might be higher than emissions propose with a constant $r_{aH}$ across the study area. It would therefore be beneficial to consider variable aerodynamic resistances and to use



models that relate canopy porosity to create maps of variability in $r_{aH}$. Further experiments should be done to determine how $r_{aH}$ and consequently the resulting $F_c$ change when using different methods of estimating $r_{aH}$.

### 4.2.2 Averaging procedure

A methodology to improve the grid averaging would be to sub-sample larger grid cells using a finer scale grid (e.g. 20 m × 20 m or less) and then averaging those finer grid cells to lower grid resolutions as done in Crawford and Christen (2014). This would help to reduce some errors at two critical moments. First, it may be possible to average out some of the extreme values within a grid cell that may be contributing to an over- or -underestimation of emissions within a grid cell due to a spatial sampling bias. Second, it offers a possibility to determine the representativeness of the grid cell sample and attribute a certainty or weight to each cell. Because the current methodology simply spatially attributes any point(s) to the grid cell in which it intersects, we do not account for the degree in which point measurements represent the spatial mean of grid cells.

### 4.2.3 Emission inventories

Several factors may account for the differences due to errors in the emission inventories. First, the emissions inventories were not based on real-time models of the data for the period of the measurement campaign. The building emissions inventory presents a challenge when comparing the grid averaged $r$ and the measured emissions because the building emissions inventory is downscaled to an hourly average from a yearly estimate. This hourly average is assumed to be constant over the course of the day, however, studies (e.g. Martani et al. (2012)) show that most building occupancy (and therefore energy use) occurs between 9:00 and 19:00, with peaks around 13:00 and 16:00. Furthermore, this does not address the fact that spatially, building energy use changes throughout the day as people go to and from work and home. Future work might attempt to quantify the spatial ebb and flow of people using a combination of surveys, census data, and methods using call detail records to derive home versus work locations as shown in Holleczek et al. (2014). Building energy use intensity might be modeled by season and diurnally based on factors such as building occupancy, building age, form, and function.

To explain differences in the traffic emissions inventory, we must account for the fact that the traffic emissions inventory was derived from spatially and temporally disaggregated samples of short-term traffic counts. As a result, the traffic emissions inventory may compound errors over time and space. Spatially, the traffic count dataset covers mostly the major roads which leaves much of the residential areas unsampled. The method described in Appendix B1 is used to map traffic count values across the residential streets to overcome the missing traffic counts, however more validation is necessary to determine whether this method is appropriate. Temporally, the traffic emissions inventory is not a real-time representation of the traffic counts during the measurement campaign. Furthermore, the traffic emissions are generated using an emissions factor that is a fleet average for the emitted $CO_2$ per liter of fuel burned. More precise estimates of emissions factor in the differences in the emissions factor by vehicle type and fuel type (Kellett et al., 2013). Last, the traffic count data does not indicate the amount of emissions from idling that occur as a result of traffic jams and thus introduces another aspect of possible uncertainty within the traffic emissions inventory, and can be substantially higher in urban contexts.





The total emissions inventory factors only building and traffic emissions and excludes other sources of emissions such as those from human, animal, and plant and soil respiration. Additional sources of $CO_2$ emissions could come from human activities such as landscaping (e.g. lawnmowers and leafblowers) and construction. For example, a study by Kellett et al. (2013) showed that, in a 1.9 km × 1.9 km study area around the "Vancouver-Sunset" tower (see Fig. 2), emissions from human
respiration and vegetation and soils can account for 8% and 5% respectively of the total emissions, respectively.

## 5    Conclusions

Several studies have measured $r$ across transects through cities (Jimenez et al., 2000; Idso et al., 2001; Henninger and Kuttler, 2007; Crawford and Christen, 2014), however no study to date has deployed multiple mobile $CO_2$ sensors simultaneously, and no study has used the measured $r$ in combination with a tower to determine emissions across a city.

A portable, mobile sensor system called the $DIYSCO_2$ was developed an tested. Five $DIYSCO_2$'s were deployed across a 12.7 $km^2$ study area over a period of 3.5 hours; the average sampling density was about 40 samples $ha^{-1}$. Of the 11.7 $km^2$ study area that could be traversed, 8.5 $km^2$ in summer and 8.2 $km^2$ in winter were sampled with $> 10$ samples per grid cell. Hence, excluding the grid cells with $< 10$ samples, the sampling density was roughly 0.5 $km^2 sensor^{-1} hr^{-1}$ over the 3.5 hour period for the 5 sensors. If it is assumed that this sampling density is appropriate for representing urban scale processes, it
would require 230 coordinated mobile sensors on predefined routes to be deployed across the entire City of Vancouver (115 $km^2$) to measure $CO_2$ emissions across the city during the same time – obviously an effort that is not realistic.

However as sensor parts will become cheaper in the future, possibilities exist to integrate mobile sensor systems into operational vehicles such as taxis (e.g. 600 in the City of Vancouver) and mobility-on-demand services (e.g. currently there are >1000 carshare vehicles in the City of Vancouver). Alternatively, the time frame could be extended and using proper data
selection, one could create composite maps from $r_{mobile}$ measured on different days under similar conditions. It would take 10 days in a coordinated effort to cover the entire City of Vancouver similar to the current transect.

A further question to be explored is whether the current number of samples ($> 10$ s) per grid cell is sufficient to represent the typical emissions in the cell given the intermittent traffic and the fact that large coherent structures are mostly responsible for mixing of pollutants out of the urban canopy layer (Salmond et al., 2005; Christen et al., 2007). Would a higher density of
points (including multiple campaign days) improve the correlation between measured and inventory emissions?

The method to map emissions based on the aerodynamic resistance approach is sensitive to the measurements that are used to derive the aerodynamic resistance of heat and requires that a number of assumptions and conditions are met, yet, the work shows that the aerodynamic resistance approach can be used reasonably on a scale of $100 \times 100$ m grid cells to derive emissions from measures of aggregated mixing ratios. The measured emissions across the study area ranged from -12 $kg\,CO_2\,ha^{-1}\,hr^{-1}$
to 225 $kg\,CO_2\,ha^{-1}\,hr^{-1}$ per grid cell, thus showing the possibility for this methodology to detect negative emissions (net uptake), where photosynthesis is greater than the combined combustion and respiration emissions.

The research presented is proof of concept for a future in which atmospheric sensing is integrated into urban mobility. We haves shown the successful development of new technology and methodology for monitoring and mapping $CO_2$ mixing ratios





and emissions in complex urban environments, at much finer scale than previously possible. Despite the simplicity of the methodology, the study demonstrated that it is possible to measure emissions across a complex landscape with a fleet of mobile sensors, an eddy-covariance tower, and the use of the aerodynamic approach to calculating emissions.

The data gained cannot be only used to map and validate emissions but could be integrated into regional efforts using
observations and inverse modelling (Newman et al. (2013)) or even with total column measurements of $CO_2$ from satellites.

Further, the concept can and should be translated to the mapping of other trace gases and air pollutants, air and surface temperature, and other environmental variables that affect human health, comfort, and safety. The development of smaller, more affordable, mobile sensor systems can facilitate new methodological approaches to monitoring the urban environment. With a fleet of mobile sensors and the methodologies for processing the derived datasets, the possibility to map and consequently
validate emissions inventories is promising, as is the derivation or real time pollution and climate data in cities.

### Appendix A:  Testing of sensor system

Several key system specifications of the $DIYSCO_2$ were evaluated during the prototyping, namely: linearrity, accuracy and drift, measurement lag time, between sampling and measurement, and the effects of inlet location on measurement variability.

### A1   Sensor precision

The accuracy of the Li-820 is ensured using a two-point calibration, usually performed in the lab using a zero-gas and a span gas in the range of assumed measurement. However, precision and linearity of the Li-820 sensor is in particular relevant in the range 400 to 500 ppm to enable comparisons between different $DIYSCO_2$'s operated simultaneously and also to properly compare $r_{mobile} - r_{tower}$.

To test the accuracy and linearity, a 6-point calibration was performed using six tanks of known mixing ratios of $CO_2$
between 399.08 and 503.77 ppm. All standard tanks have been calibrated against CDML / NOAA WMO traceable tanks with a typical error in $r$ of $< 0.1$ ppm. To perform this test, all Li-820 sensors were first left running for 2 hours to ensure significant warm up time. After the warm up period, the $DIYSCO_2$'s were connected to a calibration gas using a Union Tee connector. For each of the six gases, the calibration protocol called for an initial two minute system flush and then a recording of the values for at least 1 minute each. A minimum of 60 points per gas sample were used to calculate the average mixing ratios per
tank measured by the system. The data were recorded directly by the $DIYSCO_2$ data logger.

The Li-820 contained in the $DIYSCO_2$ showed strong linearity ($R^2$ of 0.9999) and a root mean square error (RMSE) of 0.233 ppm for the six tanks of known $CO_2$ mixing ratios. This indicates that the IRGA is operating well within its factory specifications of 1 ppm when calibrated and linearity is not a limiting factor for this type of study.





## A2  Sensor drift

Sensor accuracy and drift is assessed to determine the DIYSCO$_2$'s ability to properly resolve the variability of mixing ratios during the duration of the campaign. Sensor drift was tested over the course of 7 days, with 5 sensors drawing in air from the same point outdoors at ≈3 m in an urban context.

The RMSE between the five system at a 1-minute resolution ranged between 0.2 and 3 ppm for the seven day period and is therefore time dependent. Given that the field campaign was planned to be 3 to 3.5 hours long, the maximum drift of any sensor in any 3 hours was determined at most -0.31 ppm and 0.51 ppm relative to the mean of all 5 sensors. The drift was up to ±3.32 ppm per day for individual sensors and days.

## A3  Measurement lag time

The system measurement lag time is the time delay from when a measurement first enters the sample inlet of the system to when the signal is registered by the sensor. The DIYSCO$_2$'s measurement lag time is important to correctly attribute measurements to their geographic space.

For a given tube length and flow rate, the lag time will differ and therefore affect the system response characteristics. The values here are for a tube length of 3 m. Lab measurements were performed in which a solenoid switch was used to pass
nitrogen gas with 0 ppm $CO_2$ into the sample tube inlet while simultaneously logging the exact second in which the solenoid was triggered. To calculate the lag time value for the system, the number of seconds were counted from when the sample enters the sample tube until 50% of the change was reached.

The measurement lag time of the DIYSCO$_2$ system was determined to be 18.2 s. It took on average 16 seconds for the sample to travel from the inlet to the IRGA and 2.2 seconds for the IRGA to register 50% of the step change. We consequently
used a value of 18 seconds in the post processing to shift the GPS and observed $r_{mobile}$ time series to properly attribute measurements spatially to locations.

## A4  Effects of inlet location

Two tests were performed to examine possible sampling biases due to different sample inlet locations on a vehicle. First, a test was done with five DIYSCO$_2$ in the same vehicle, where all the inlet tubes were bundled together measuring at the same
location of the vehicle (referred to as "Grouped Inlet Test"). A second test was done with each of the inlet tubes located at different locations on the same vehicle (referred to as "Ungrouped Inlet Test"). Locations tested were all at 2 m height: One each above the driver's side front, driver's side back, passenger side front, and passenger side back window.

Both test were performed in the City of Vancouver using a Toyota Tacoma Truck along a route with traffic volumes ranging from 300 to 850 vehicles per hour. In areas with a well-mixed atmosphere and on roads with little traffic, the DIYSCO$_2$ systems
for the grouped inlet test showed a range within ±0.5 ppm of the mean all five sensors for 1 second data. For the ungrouped inlet test under those same conditions, the accuracy deteriorated to ±5 ppm of the mean. With observations of higher $CO_2$ mixing ratios, the standard deviation between all five of the DIYSCO$_2$ locations increases for the 1 s data. This is the case for



both the grouped and ungrouped inlet tests. With inlets grouped together, 48.9%, 81.16%, and 90.14% of the one second data have an error within 5, 15, and 25 ppm. While this indicates that more than half of the 1-s data measured by the sensors are within 5 ppm of each other, the test also shows that we can expect a majority of the data (>88.85%) to have errors up to 15 ppm depending on where on the car the inlet is mounted. When examining the error of the observed values for the 1 min data,

we observed that 86.3% and 98.63% of the data have an error within 5 and 25 ppm.

**Appendix B: Emissions inventories**

This Appendix described the derivation of the independent building and traffic emissions inventory that were compared against the measured $CO_2$ emissions.

**B1    Traffic emissions inventory**

The fine-scale gridded traffic emissions inventory was based on hourly averaged directional traffic count data from 2008 - 2013 provided by the City of Vancouver (City of Vancouver, 2015).

     For each hour of the day, traffic counts were spatially attributed to the Open Street Map road network. The City of Vancouver provides traffic counts collected from pneumatic road tubes which are attributed to an approximate address of where the traffic counters were located. The traffic counts do not distinguish between different vehicle classes and are aggregated to the street

level, meaning that, for this analysis, the traffic counts did not take into account the direction of travel.

     The City also provides a geospatial representation of the locations of the traffic counters with the address, but without the count data attached. The geospatial data were merged with the count data. However, because spatial traffic counts do not align with the OSM road network, the centroids of the spatial traffic count data were computed and then "snapped" to the OSM road network. Before joining the traffic count data by the matching locations of the two datasets, the OSM road network was split

into segments using the 50 m × 50 m vector grid. A small (0.5 m) buffer was applied to the traffic count centroids to ensure that they spatially match onto the OSM road network and then were merged to the OSM dataset.

     An algorithm was used to match the street names in the traffic count dataset to those in the OSM street network. Manual mapping of traffic counts was necessary to attribute traffic counts to streets that were not sampled in the traffic counts. A rule of proximity and local understanding of the traffic patterns for each of the streets was used to manually map the traffic counts

to the unsampled streets. Using the OSM street classifications, traffic counts for paths unnavigable by vehicles were given a value of "0" traffic counts, namely "steps", "trail", "footpath", and "service". Lastly, the traffic counts for forked roads in the dataset which would have doubled the count for a particular street were divided in half.

     With a complete model of the traffic counts for the transect, it was then possible to generate a gridded traffic emissions inventory map of $CO_2$ (now referred to as "traffic emissions inventory"). The length of each of the street segments which had

been split in the earlier steps were calculated and then summed up per 50 m, 100 m, 200 m, and 400 m grid cell. Next, the length of navigable roads per grid cell were multiplied by the hourly traffic counts along each road, resulting in an estimate of total distance of vehicle traveled per grid cell. Each grid cell's hourly travel distance was then multiplied by the NRCAN





fleet standard fuel comsumption (Natural Resources Canada, 2014) for urban driving ($12.9\,\ell\,100\,\mathrm{km}^{-1}$) and after by a $CO_2$ emissions factor ($2.175$ kg $\ell^{-1}$ fuel burned) (Environment, 2014) to generate the traffic emissions estimate map of $CO_2$. In this study, the traffic count data provided by the City of Vancouver is averaged across all of the years that the traffic count data have been collected. The data are then scaled by a factor 0.9985 and 1.0216 to reflect the seasonally changing relative traffic

volumes for March and May based on automatic and continuous highway counts (weekday only) at 5 locations throughout Metro Vancouver.

## B2  Building emissions inventory

The fine-scale gridded building emission inventory was developed in previous research and is documented in detail in van der Laan (2011). It integrates Light Detection and Ranging (LiDAR) data, building simulation software and a building typology

database to model $CO_2$ emissions attributed to building energy use; The original building emissions inventory is on a per-building scale in carbon dioxide equivalent ($CO_2e$, reported in kg $CO_2e$ $yr^{-1}$). In this research, it is assumed that $CO_2e$ and $CO_2$ are the same for building heating systems. This is then then converted to a 1m raster using building footprints derived from LiDAR and property permeters. The 1 m raster was then averaged to the 50 m, 100 m, 200 m, and 400 m vector grids and scaled to their estimated hourly values for both campaigns.

Because the inventory by van der Laan (2011) reports annual estimates (in kg $CO_2e$ $m^{-2}$ $yr^{-1}$), a scaling factor based on monthly city emissions inventory was used in this study to account for the winter and summer building emissions fraction. In the month of March and May, the building emissions for a sample of the City of Vancouver was estimated to be 99.85% and 63.63% of the annual average building emissions (reported in (Christen et al., 2011)). The final building emissions inventory were reported in kg $CO_2$ $ha^{-1}$ $hr^{-1}$. In this case, it is assumed that the building emissions are constant over the course of the

day.

Each grid cell of the total emissions inventory is simply the sum of the building emissions inventory and the traffic emissions inventory in kg $CO_2$ $ha^{-1}$ $hr^{-1}$. Other emission processes such as human respiration or biological processes are not considered in the inventory.

## Appendix C:  Effect of grid size

In addition to the $100 \times 100$ m grid, the raw data points were also gridded to 50 m, 200 m, and 400 m vector grids for both the winter and summer campaigns to explore the sensitivity of choosing different grid sizes.

## C1  Effects on spatially averaged mixing ratios

Changes in grid size affected the study area mean $r_{\mathrm{mobile}}$ by 6.1 ppm in the summer and only 1.1 ppm in the winter. Table 4 summarizes the statistics for different grid cell sizes. The grid maximum values for the 50 m, 100 m, 200 m, and 400 m grids

were 529.8, 518.0, 488.2, and 447.7 ppm respectively for the summer and 643.1, 560.5, 529.4, and 492.5 ppm respectively for the winter.





**Table 4.** Summary data of the measured mixing ratios for all grid sizes for the summer and winter campaigns. The table shows the mean, minimum, median, maximum $CO_2$ mixing ratio $r_{\text{mobile}}$ for the gridded data.

| Grid Size | Min (ppm) | Median (ppm) | Mean (ppm) | Max (ppm) |
|---|---|---|---|---|
| **Summer** | | | | |
| 50m | 393.1 | 409.4 | 417.3 | 529.8 |
| 100m | 393.1 | 410.0 | 417.9 | 518.0 |
| 200m | 397.0 | 412.9 | 419.6 | 488.2 |
| 400m | 399.6 | 417.5 | 419.0 | 447.7 |
| **Winter** | | | | |
| 50m | 408.4 | 434.5 | 442.6 | 643.1 |
| 100m | 408.4 | 435.0 | 442.5 | 560.5 |
| 200m | 408.4 | 436.8 | 443.7 | 529.4 |
| 400m | 420.5 | 441.9 | 443.2 | 492.5 |

The highest grid maximums were observed in the 50 m grid size. This is expected because the most extreme $r_{\text{mobile}}$ are spatially averaged out by larger grid cell sizes.

## C2 Effects on spatially averaged emissions

In the summer campaign, the differences between the measured emissions and the inventory emissions increases as the grid size increases (Tab. 5). The least amount of difference is seen in the 50 m grid at $6.88 \, \text{kg} \, CO_2 \, \text{ha}^{-1} \, \text{hr}^{-1}$. In the winter campaign, the differences between the measured and inventory emissions are smallest in the 100 m ($2.84 \, \text{kg} \, CO_2 \, \text{ha}^{-1} \, \text{hr}^{-1}$) and 200 m ($0.9 \, \text{kg} \, CO_2 \, \text{ha}^{-1} \, \text{hr}^{-1}$) grid sizes and are greatest in the 50 m grid size at $7.8 \, \text{kg} \, CO_2 \, \text{ha}^{-1} \, \text{hr}^{-1}$.

In both campaigns, the spatial error (expressed RMSE) between measurements and inventory decreases as grid sizes become coarser. In the summer campaign 80.05%, 86.71%, 85.31%, and 95.45% of the cells have measured emissions that are within a factor of $\pm$ 10 of the total emissions inventory for the 50 m, 100 m, 200 m, and 400 m grids, respectively. In the winter campaign, 91.16%, 93.74%, 94.20%, and 100% of the cells have measured emissions within $\pm$ 10 of the total emissions inventory.

For the winter campaign, we observe as grid size increases, the mean bias, i.e. differences between the mean measured emissions and the mean inventory emissions decreases, presumably because more sampling points mean we average out random errors in individual cells. The best match is found at 200 m resolution. Of course this is very sensitive to the calculated aerodynamic resistance and should not be interpreted as a generality.

For the summer campaign, however, there is an increasing difference between the mean measured emissions and the mean total emissions. This may be best explained by the bias towards roads in the sampling methodology. In the summer campaign,



**Table 5.** Mean measured emissions versus mean inventory emissions for the winter and summer campaigns

| Grid size | Measured emissions ($\mathrm{kg\,CO_2\,ha^{-1}\,hr^{-1}}$) | Inventory emissions ($\mathrm{kg\,CO_2\,ha^{-1}\,hr^{-1}}$) | Relative difference | RMSE ($\mathrm{kg\,CO_2\,ha^{-1}\,hr^{-1}}$) |
|---|---|---|---|---|
| **Summer** | | | | |
| 50m | 34.06 | 27.18 | +29% | 32.54 |
| 100m | 35.11 | 22.06 | +59% | 27.91 |
| 200m | 38.30 | 19.73 | +94% | 29.01 |
| 400m | 37.26 | 15.27 | +144% | 28.57 |
| **Winter** | | | | |
| 50m | 25.67 | 33.47 | -23% | 34.23 |
| 100m | 25.92 | 28.76 | -10% | 25.39 |
| 200m | 27.21 | 26.31 | +3% | 19.58 |
| 400m | 26.60 | 23.33 | +14% | 17.71 |

the dominant source are vehicles constrained to roads. The difference between the average measured emissions and the total emissions inventory is relatively small for the 50 m grid because the measurements are made mostly along roads and therefore do not include traffic-free areas such as in the backyards of homes and within large street blocks which can have significantly lower concentration of traffic-related pollutants (Weber and Weber, 2008). As a result, when comparing the average measured emissions to the average of the total emissions inventories for the 100 m, 200 m, and 400 m grids, we see that a sampling bias becomes more apparent. The 50 m grid cell size is a more appropriate resolution for griding the point measurements collected using this methodology when traffic emissions dominate. Additional sampling along alleys and laneways and more representative sampling using alternative mobility options such as bikes or autonomous flying vehicles may help to improve the relationship between measured emissions and the emissions inventory when griding at coarser resolutions.

*Acknowledgements.* This funding was supported through an NSERC Discovery Grant ("Direct measurement of greenhouse gas exchange in urban ecosystems", A. Christen). Sensor development and tower infrastructure were funded in part through the Canada Foundation for Innovation (Grants 17141 and 33600). Scholarship to J. K. Lee was provided through NSERC CREATE and through the "Mozilla Science Lab", . Experiment vehicles were sponsored by "moovel lab", Stuttgart, Germany. We thank A. Black, R. Kellett, S. Lapsky, L. Lavkulich (all UBC) for their guidance, support, and help.



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
