# Peer review of "A mobile sensor network to map carbon dioxide emissions in urban environments"

_Atmospheric Measurement Techniques, 2016_

## Referee Comment (RC1) · Dr. Velasco (Referee) · 11 Jul 2016

Lee et al. propose a measuring system based on a mobile network of sensors to validate estimations of carbon dioxide ($CO_2$) emissions at fine spatial resolution (i.e. grid cells of 100 m2) in urban areas. They built and tested an initial network of five sensors as a proof-of-concept. The sensors are basically formed by a commercial $CO_2$ monitor and a Global Positioning System (GPS) connected to a low-cost controlling board and placed in a special box for being carried on mobile platforms such as cars and bicycles.

The $CO_2$ concentration data collected by the sensors along roads, streets and paths of the city are used to estimate emissions applying an aerodynamic resistance approach and sensible heat flux data obtained from an eddy covariance flux tower located within

CC-BY license logo

a sector of the city monitored. As a proxy of the aerodynamic resistance of $CO_2$, they used the aerodynamic resistance of sensible heat under the strong assumption of equivalence between both.

The testing results showed the capability of the mobile network to complement eddy covariance flux measurements and validate emission estimates based on activity data and emission factors. The proposed monitoring system represents a new tool to solve the puzzle of the greenhouse gas emissions at city scale. As any other approach, it has advantages and limitations. Both are discussed in the manuscript.

The description, discussion and validation of the proposed approach fit well within the scope of Atmos. Meas. Tech. This work represents, in general, a valuable contribution to the ongoing efforts to quantify urban emission in a way to support policies for climate change mitigation.

The technical issues to be addressed are minor. However, this reviewer cannot recommend the immediate publication of the manuscript because of severe problems in the writing. The structure of the manuscript is appropriate, but the writing is not good enough for a scientific paper. A number of sentences are repetitive and others confusing. The manuscript needs a comprehensive editorial revision to be considered for publication.

After the technical comments, a number of editorial suggestions are listed for the first ten pages of the manuscript. This reviewer expects they can provide some insight on how to fix the writing in general.

Technical comments (Page/Line)

Since the mobile $CO_2$ measurements were conducted along roads and streets, the approach is biased to traffic emissions. How this issue could be addressed, in particular for other trace gases, such as methane, whose origin relies in source emissions other than traffic? Please emphasise this issue even more.

Data-driven models in combination with databases of urban parameters, such as land-use, building characteristics, population density, vehicular traffic, etc. at fine spatial scale may help to identify grid cells of similar characteristics and better estimate their emission. They could be used to extrapolate emissions for those cells not included in the sampling transects, as well as to determine the ideal number of sampling points per cell. If those databases are available, the approach proposed here may improve significantly its performance. See Moosavi et al., Atmos. Meas. Tech. 8, 3563-3575, 2015.

10/21-24 Was the stationarity criteria for the eddy covariance flux data used to remove suspicious periods during the mobile measurements? I mean, if a flux measurement period did not meet the stationarity criteria, were the mobile data collected during the same period also discarded?

Table 2 & 3. Are statistically different ($p \leq 0.05$) the concentrations/emissions recorded/estimated between the grid cells of each neighborhood and between neighborhoods?

21/4-6 How do you explain that the measured emissions were in average higher in summer than in winter, contrary to the estimated emissions by bottom-up approaches?

21/23 Indeed, the approach lead to realistic and consistent results in average when evaluated at neighbourhood scale, but not at fine scale (i.e. grid cells).

22/10-14 The comparison should be restricted to the same periods of the day on weekdays and same climatological seasons.

Editorial suggestions (Page/Line)

1/11 Why the use of quotation marks?

1/17 Eighty seven percent (summer) and 94% . . .

2/3 Define directly and indirectly emissions. Not all readers might be familiar with these

terms.

2/10 It depends on the latitude (e.g., Velasco et al., Landsc. Urban Plan. 148, 99-107, 2016).

2/15 Avoid acronyms to start a sentence.

2/16 Does it sound better quick or fast instead of agile?

2/24 The research goal of this study/work is to develop . . .

2/25-27 Rewrite. For example: Data from a network of mobile sensors and an eddy covariance flux tower combined with an aerodynamic approach are used to calculate and map . . ..

2/28 Mobile measurements have been used . . .

2/30 mobile monitoring methods rely on a . . .

2/31 – 3/2 These two sentences are repetitive, merge them with the previous one.

3/3 I would say interest instead of success.

3/4 Top-down data mining?

3/8 Not all readers might be familiar with "bottom-up approaches".

3/16 "Autonomous flying vehicles" sounds like science fiction.

3/3 – 3/17. The whole paragraph needs to be rewritten. This reviewer does not consider necessary the discussion on the use of open-source microcontrollers in combination with cell-phones as a proxy to map environmental parameters. This work proposes the use of instrumentation specifically designed for measuring $CO_2$.

3/20 This study investigates the feasibility for mapping greenhouse gas emissions, specifically $CO_2$ . . .

3/22 Replace "car sharing platforms . . . or random vehicles" by "mobile platforms".

3/24-28 These four bullets sound more like the manuscript's structure rather than the objectives. Rewrite them in one paragraph.

4/1-2 Never leave titles/subtitles without text. Don't use uncommon acronyms for a title. All acronyms need to be previously defined in the text.

4/4 This sentence is repetitive.

4/5 What about "passenger" instead of "various"?

4/7 Define limited time-scale and fine resolution. One, two five hours? Grid cells of 50, 100, 500 m2?

4/9-10 . . . have been stationary or mounted in specialized vehicles . . .

4/13-16. Rewrite the whole paragraph. For example: Carbon dioxide analysers based on infrared detection (Licor . . .) were coupled with microcontrollers (Arduiono . . .), etc.

4/14 12.23 × 15.25 × 7.62 cm

4/16 No need ". . . monitoring applications including agriculture."

4/19 Define GPS.

4/27 Material of the tube?

5/Panel b. Which is the height of the sampling line over the vehicle's roof?

5/Figure 1 legend. It is clear that both panels show photos of the CO2 system. No need of indicating that during operation the system is enclosed in the case neither that the used vehicle was a car-shared one.

6/10 Indicate that in Canada, the driver position is at the left side.

6/19-26 Rewrite the whole paragraph.

6/20 Provide locations using latitude and longitude in degrees.

6/27 Flux tower measurements.

6/28 Eddy covariance flux tower . . .

6/28 No need to include the Fluxnet ID of the flux tower. The manuscript never makes reference to Fluxnet.

6/28 – 8/9 This paragraph needs to be rewritten. Indicate since then the flux tower has been working. Provide a reference to a comprehensive description of the tower.

7/Figure 2 legend. Avoid repeating information already given in the text (e.g., 12.7 × 1 km). Try to use active sentences as much as possible (e.g., The location of the flux tower is marked by . . . instead of Shown are also . . .).

7/Figure 2 legend. Crowford and Christen, 2014 is a very good paper, but I do not consider necessary to include it here.

8/11-15 Rewrite the whole paragraph. For example: Two fields campaigns took place, the first on . . . during the summer and when trees leaves are in full, while the second on . . ., covering the heating season. Sampling was conducted from 10:00 – 13:30 h, when vehicular traffic and meteorological conditions are relatively constant.

8/16. Remove this sentence.

8/17 Replace predefined by designed.

8/16-19 Rewrite these two sentences.

8/23 Should be bicycle instead of bike?

9/3 Describe briefly such filtering methods.

9/20 Data from the eddy covariance tower are . . .

10/Eq. 3 So many conversion factors at the beginning of the equation are confusing. Merge them in only one factor and explain its meaning in the text.

10/15 Remove "in the two measurement campaigns". It is obvious.

10/26-30. Make shorter this sentence. For example: This assumption is supported by a previous study in which no storage flux was observed during daytime for this particular site (Crawford and Christen, 2014).

11/1-3 Any reference on the Reynolds analogy?

11/11 Better indicate the percent of excluded or included readings.

14/Table 2. Indicate which those climate zones are.

23/17 . . . between 9:00 and 19:00 h . . . This might be true for office buildings, but not for residential buildings. 23/30 . . . the traffic count data do not indicate . . .

24/7-9 Do not list what others have made (you did it already in the introduction), better say that this study was the first in combining . . .. to evaluate CO2 emissions at fine scale.

25/19 Replace six tanks by six standard gases

27/12 . . . to the Open Street Map (OSM) . . .

---

## Referee Comment (RC2) · Anonymous Referee #2 · 20 Nov 2016

Review of Lee at al. "A mobile sensor network to map CO2 emissions in urban environments" The paper presents a detailed methodology for measuring spatial CO2 emission in an urban landscape using low-cost sensor system deployed on vehicles in urban areas. Methodology for the estimation of CO2 fluxes from urban areas is compared with EC approaches measured using a traditional flux tower over Vancouver. This is a very detailed description of the approach and the validation against established methods and one can see the extreme usability of such a system. The paper is well within the remit of amt and the authors have a novel approach. This type of work is needed to address the major challenges of addressing the study of the urban atmosphere, not least its spatial heterogeneity. The authors discuss the various advantages and disadvantages of their approach. I am very happy at the level detail shown by the authors especial with the design and construction of the DIYCO2 system. I would

suggest the authors put the appendix in the supplementary information section of the article.

---

## Short Comment (SC1) · 11 Dec 2016

I would like to add some comments to help strengthen the description of the DIYSCO2 system:

Appendix A1 is titled 'sensor precision', the first paragraph says accuracy is ensured by calibration but precision and linearity need to be tested, then the second paragraph says accuracy and linearity are tested, and the final paragraph concludes about linearity – please sort out the usage of each of these terms.

6/5 and Appendix A1: This at first glance suggests impressive linearity, however there is not enough information to evaluate this claim. As an extreme example, if five of the six tanks had a mixing ratio of 399.08 ppm and the sixth was at 503.77 ppm you could get a linear R2 of 0.9999 even with a nonlinear sensor - please state the values of each

tank. It would also be helpful to know how the standard tanks were calibrated against CDML / NOAA traceable tanks as was stated.

Appendix A3 states a time response of 2.2 seconds for a 50% step change within the IRGA. This indicates that the effective number of samples is less than the 1 Hz that is stated, and could have an implication for the effective sampling density of measurements in the city. Please include a statement about whether the time response of the sensor being > 1 s affects the results.

Appendix A4 This is an interesting test, but the second paragraph is somewhat difficult to read and several things need to be clarified.

- What exactly is the difference between the tests in p26/L30 where good agreement was seen between the grouped inlets, and p27/L1 where the grouped inlets had relatively large spread? By higher CO2 mixing ratios, does this mean this second results were for an area with direct traffic emissions?

- What about the results of the ungrouped test? Perhaps this is why p27/L3 says 88.85%, while p27/L1 says 81.16% were within 15 ppm?

- p27/L1 implies that slightly less than half of the 1-s data is within 5 ppm, contradicting the next sentence. I also disagree with the use of 'accuracy' and 'error' here, since there is no standard to compare against other than the mean value, and would recommend spread or variability instead.

If the authors move the appendices to a supplement per reviewer 2's comment, I would suggest retaining Appendix A since this would give a disproportionally large amount of space to the field results and comparison to inventory, in comparison to description of DIYSCO2 and the emission calculation methodology.

2/15: This says accuracy, but the value given in parentheses is the 1-s noise specified in the instrument datasheet, not its accuracy. The Li-820 manual gives accuracy specifications, based on mixing ratio range and cell pressure, as a percentage (%) of

reading

8/29: "In-situ calibration". This is a good thing to do, but should be called a comparison not a calibration since there is no standard used to calibrate against. Can also a comment also be made about whether this provided useful information? If this means parking five vehicles adjacent to one another, and given the variability described in Appendix A4, I imagine it might be difficult to detect drifts this way both for individual sensors and for the ensemble of sensors.

12/5: What is the meaning of "sample" in this section. Does one sample correspond to one 1 Hz measurement? If so, there should be some discussion about vehicle speed since that will affect the spacing of measurements.

Other:

* Spell out IRGA at first use

* p2/L14 and p8/L5: NB -> NE

* p9/L2: gridding
* * *

---

## Author Comment (AC1) · 25 Jan 2017

Final author's response to all reviewer and public comments.

Note: The attached 'supplement' contains the manuscript with all changes tracked / highlighted for reference.

**Reviewer 1: Erik Velasco**

\* Lee et al. propose a measuring system based on a mobile network of sensors to validate estimations of carbon dioxide (CO2) emissions at fine spatial resolution (i.e. grid cells of 100 m2) in urban areas. They built and tested an initial network of five sensors as a proof-of-concept. The sensors are basically formed by a commercial CO2 monitor and a Global Positioning System (GPS) connected to a low-cost controlling

board and placed in a special box for being carried on mobile platforms such as cars and bicycles.

The $CO_2$ concentration data collected by the sensors along roads, streets and paths of the city are used to estimate emissions applying an aerodynamic resistance approach and sensible heat flux data obtained from an eddy covariance flux tower located within a sector of the city monitored. As a proxy of the aerodynamic resistance of $CO_2$, they used the aerodynamic resistance of sensible heat under the strong assumption of equivalence between both.

The testing results showed the capability of the mobile network to complement eddy covariance flux measurements and validate emission estimates based on activity data and emission factors. The proposed monitoring system represents a new tool to solve the puzzle of the greenhouse gas emissions at city scale.

As any other approach, it has advantages and limitations. Both are discussed in the manuscript. The description, discussion and validation of the proposed approach fit well within the scope of Atmos. Meas. Tech. This work represents, in general, a valuable contribution to the ongoing efforts to quantify urban emission in a way to support policies for climate change mitigation. The technical issues to be addressed are minor.

- RESPONSE: We appreciate your accurate summary and assessment of the work's impact. Thank you.

However, this reviewer cannot recommend the immediate publication of the manuscript because of severe problems in the writing. The structure of the manuscript is appropriate, but the writing is not good enough for a scientific paper. A number of sentences are repetitive and others confusing. The manuscript needs a comprehensive editorial revision to be considered for publication.

- RESPONSE: We revised the writing in the entire manuscript, removed repetitions and ensured scientific style. All substantial changes are documented below and/or in the

accompanied document "tracked-changes.pdf"

After the technical comments, a number of editorial suggestions are listed for the first ten pages of the manuscript. This reviewer expects they can provide some insight on how to fix the writing in general.

**Technical comments:**

* Since the mobile CO2 measurements were conducted along roads and streets, the approach is biased to traffic emissions. How this issue could be addressed, in particular for other trace gases, such as methane, whose origin relies in source emissions other than traffic? Please emphasise this issue even more.

- RESPONSE: Addressed. It is correct that due to the nature of the mobile sensing approach on cars and bicycles, measurements are primarily taken along roads and streets with traffic. However, in the planning of the routes, we also defined many segments along laneways (alley ways behind houses and buildings) - wherever and as often as possible. Laneways have very limited traffic (access to garages, garbage disposal trucks). Measurements in laneways are an attempt to better represent the area. We also drove into cemeteries and parks and used a bicycle to access pedestrian-only pathways in parks. The current approach has been tested only for emissions from vehicles and emissions at roof-level (e.g. emissions from furnaces released though chimneys that then are mixed down). We agree, that this methodology might not be transferrable to emissions whose origin are other than traffic. We therefore added a sentence of caution to the conclusions: "However, due to the assumption that sources are in the canopy layer where sensors operate, the proposed methodology is not necessarily transferable to emissions whose sources are not well represented such as fugitive natural gas emissions (methane) or volatile organic compounds or large industrial sources (tall stacks)."

* Data-driven models in combination with databases of urban parameters, such as land- use, building characteristics, population density, vehicular traffic, etc. at fine spa-

tial scale may help to identify grid cells of similar characteristics and better estimate their emission... (See Moosavi et al., Atmos. Meas. Tech. 8, 3563-3575, 2015)

- RESPONSE: This is an interesting idea, and could - in the post processing - be further used to achieve more stable emission estimates. We added the a sentence and a reference to Moosavi et al. 2015 to the 'future improvements' section (just before conclusions). We did not apply this approach as it would make our comparison to the emission inventory not independent. The emissions inventory for building emissions depends on detailed urban form information, the emissions inventory for vehicular traffic relies on traffic counts. In the manuscript we independently compare emission inventory and measured data, which would not be possible with the proposed data-fusion approach.

* 10/21-24 Was the stationarity criteria for the eddy covariance flux data used to remove suspicious periods during the mobile measurements? I mean, if a flux measurement period did not meet the stationarity criteria, were the mobile data collected during the same period also discarded?

- RESPONSE: All fluxes were processed according to the guidelines defined in the "Environmental Prediction in Canadian Cities Network" following Crawford et al. (2011) which included spike removal, tests for statistical behaviour, tests against other systems [Crawford B., Christen, A., Ketler R. (2009): 'Processing and quality control procedures'. EPiCC Technical Report No. 1, 11pp. http://hdl.handle.net/2429/45079]. There was no additional processing or stationarity test run on top of this. Individual values were all 'valid'.

* Table 2 & 3. Are statistically different (p $\leq$ 0.05) the concentrations/emissions recorded/estimated between the grid cells of each neighborhood and between neighborhoods?

- RESPONSE: A student's t-test was run on the gridded values for each combination of neighborhood. In the summer campaign, all neighborhoods were significantly differ-
ent from each other neighborhoods (p ≤ 0.05), except for the combinations between "Stanley park", "Sunset - Victoria-Fraserview" and "Kensington-Cedar Cottage - Riley Park", and the combination "Stanley Park" vs. "West End" which were not statistically significant. In the winter campaign, the same pattern was found, but in addition, the combination between "Fairview - Mount Pleasant" and "West End" was insignificant, while the combination "Stanley Park" vs. "West End" was significant. The table R1 summarizes the statistical tests.

\* 21/4-6 How do you explain that the measured emissions were in average higher in summer than in winter, contrary to the estimated emissions by bottom-up approaches?

- RESPONSE: This is an interesting question, that has different possible answers as discussed in the conclusions. Although we assign March 18, 2016 as "winter", it was really at the end of the season, and home heating sources were certainly less than they would be in the middle of winter. The daily average temperature of 6.1°C, however, is not different from the average of 6.9°C for March (1981-2010). A minor complication is that March 18, 2016 is during school holidays, which might have caused less traffic than during school days, but does not explain the significant mismatch. In the model, traffic counts were used irrespective of whether schools were open or closed (but only mid-week traffic counts were selected). It might also show the sensitivity of the aerodynamic resistance.

\* 21/23 Indeed, the approach lead to realistic and consistent results in average when evaluated at neighbourhood scale, but not at fine scale (i.e. grid cells)

- RESPONSE: Agree. The approach lead to realistic and consistent results only in average when evaluated at neighbourhood scale, not at the individual grid-cell. This was already stated in the conclusions, but has been updated.

\* 22/10-14 The comparison should be restricted to the same periods of the day on week-days and same climatological seasons

- RESPONSE: Although the model presented in Christen et al. (2011) separates different months (Table 2 in Christen et al., 2011), it does not separate between weekdays. We replaced the annual comparison with a comparison to the respective months of the year as follows: "Data can be compared to an independent previous study by Christen et al. (2011) that measured and modelled emissions within a 1.9 x 1.9 km study area centered on the "Vancouver-Sunset" tower (see Fig. 1). In the 1.9 x 1.9 km area, emissions were modelled 34.0 kg $CO_2$ ha-1 hr-1 and measured emissions by eddy covariance were 30.8 kg $CO_2$ ha-1 hr-1. The current study estimates emissions for the "Sunset / Victoria-Fraserview" neighborhood (that is larger than the area in Christen et al. (2011), Fig. 1) for March 18 (winter) as only 16.8 kg $CO_2$ ha-1 hr-1. For the month of May, Christen et al. (2011) report modelled emissions of 26.9 kg $CO_2$ ha-1 hr-1 and measured emissions of 26.0 kg $CO_2$ ha-1 hr-1. The current study matches extremely well here, with emissions for "Sunset / Victoria-Fraserview" on May 28 (summer) of 26.5 kg $CO_2$ ha-1 hr-1. Note that not only the spatial extent, but also the time scales of the two studies disagree. Christen et al. (2011) report monthly 24-hr emissions for the years 2008 - 2010, while the current study is restricted to weekdays between 10:00 and 13:30 on the two given dates."

**Editorial suggestions:**

* 1/11 Why the use of quotation marks?

- RESPONSE: Done. We removed quotation marks. Also in later instances, the term "measured" is used without quotation marks.

* 1/17 Eighty seven percent (summer) and 94%

- RESPONSE: Done. Changed as proposed. 87% is spelled out as "Eighty seven percent".

* 2/3 Define directly and indirectly emissions. Not all readers might be familiar with these

[Figure]

- RESPONSE: Direct emissions are emission that occur within the area investigated (e.g. administrative unit, grid cell), due to activities within the same area. Indirect emissions are emissions that occur outside the area investigated, but are linked to activities within the area (e.g. local electricity useage links to power plant emissions elsewhere). In the context of cities, the direct emissions refer to emissions that occur within the built-up area, while the indirect ones are the emissions needed to sustain cities or due to transport outside cities. We feel introducing the context of direct and indirect emissions is not essential for the current work, and a detailed definition would unnecessarily complicate the introduction paragraph. Consequently, we rewrote and simplified the sentence without using the terms "directly" and "indirectly" as: 'On the global scale, urban areas are responsible for up to 80% of the total anthropogenic CO2 emissions footprint (Satterthwaite, 2008)'. At the end of the paragraph, we clarified then '..., although a large fraction of the emissions related to the resource chains that sustain cities does not occur within the built-up area, but rather is emitted elsewhere.'

* 2/10 It depends on the latitude (e.g., Velasco et al., Landsc. Urban Plan. 148, 99-107, 2016)

- RESPONSE: Done. Thank you for clarifying this aspect. We edited the as follows and added the reference: "Overall, fossil fuel sources dominate CO2 fluxes in cities. The sequestration of CO2 by urban vegetation in most cities is very limited (Velasco et al., 2016). However, the rate of CO2 uptake by photosynthesis at a given time, can be relevant and is measurable in highly vegetated cities during the daytime in the growing season (Peters et al., 2012, Weissert et al., 2014)."

* 2/15 Avoid acronyms to start a sentence.

- RESPONSE: Done. Changed sentence as follows: "According to IPCC (2014), the urban scale has the highest potential for fast, efficient, and sustained implementation of mitigation efforts."

* 2/16 Does it sound better quick or fast instead of agile?

- RESPONSE: Done. Changed to "fast, efficient, and sustained"

\* 2/24 The research goal of this study/work is to develop...

- RESPONSE: Done. Changed to "The research goal of this study is to develop, ..."

\* 2/25-27 Rewrite. For example: Data from a network of mobile sensors and an eddy covariance flux tower combined with an aerodynamic approach are used to calculate and map...

- RESPONSE: Done. Changed as proposed: "Data from a network of mobile sensors and an eddy covariance flux tower combined with an aerodynamic approach are used to calculate and map emissions at fine scales (blocks to neighborhoods) in cities."

\* 2/28 Mobile measurements have been used...

- RESPONSE: Done. Changed as proposed to "Mobile measurements have been used..."

\* 2/30 mobile monitoring methods rely on a...

- RESPONSE: Done. Changed as follows "These studies relied on single, ..."

\* 2/31 – 3/2 These two sentences are repetitive, merge them with the previous one.

- RESPONSE: Done. In an effort to reduce redundancy we rewrote as follows: "Because trace gas analyzer systems for greenhouse gases are still bulky (e.g. Tao et al. 2015), past mobile mapping studies utilized specialized research vehicles (Bukowiecki et al., 2002, Elen et al., 2013, Crawford and Christen, 2014). While these vehicles have the advantage that they can be equipped with additional components such as calibration tanks or computers, the complexity of such systems does not allow for easy deployment on standard and flexible modes of transport."

\* 3/3 I would say interest instead of success.

- RESPONSE: Agree. Changed to "There is increasing interest to develop innovative.."

\* 3/4 Top-down data mining?

RESPONSE: Done. We deleted this sentence to remove also confusion with bottom-up approached (see next comment).

\* 3/8 Not all readers might be familiar with "bottom-up approaches".

RESPONSE: Done. We agree that not all readers might be familiar with "bottom up approaches" and have deleted the sentence.

\* 3/16 "Autonomous flying vehicles" sounds like science fiction

- RESPONSE: Done. Changed to "drones".

\* 3/3 – 3/17. The whole paragraph needs to be rewritten. This reviewer does not consider necessary the discussion on the use of open-source microcontrollers in combination with cell-phones as a proxy to map environmental parameters. This work proposes the use of instrumentation specifically designed for measuring CO2.

RESPONSE: Agree. We removed the discussion of cell-phones. Due to lack of use of distributed CO2 sensors, we showcase two examples of low-cost networks for air and surface temperature, and then dramatically shortened the paragraph as follows "There is increasing interest to develop innovative methods for monitoring urban climate and air pollution using low-cost distributed sensor networks. For example, Meier et al. (2015) used sensor data from a commercial consumer-grade weather station network to examine fine-scale urban heat island effects in the city of Berlin. In another example, Chapman et al. (2015) developed a road sensor network to monitor road surface temperatures to optimally salt roads during the winter months in Birmingham. Given this growing interest in distributed sensing systems and the advances in related technologies, could there be new opportunities for the fine-scale mapping of CO2 emissions in cities?"

\* 3/20 This study investigates the feasibility for mapping greenhouse gas emissions, specifically CO2...

RESPONSE: Done. Changed sentence as follows: "This study investigates whether it is feasible to map greenhouse gas emissions, specifically $CO_2$ at a spatial resolution of neighborhoods / blocks across the city with a portable network of mobile sensors that can be routinely implemented on various mobile platforms."

\* 3/22 Replace "car sharing platforms... or random vehicles" by "mobile platforms".

- RESPONSE: Done. Changed to: "implemented on various mobile platforms."

\* 3/24-28 These four bullets sound more like the manuscript's structure rather than the objectives. Rewrite them in one paragraph.

RESPONSE: Not changed. We argue that the organization into four sequential objectives (1-4) is useful to understand the work done, and does not impose a particular manuscript structure. Those are technical objectives that were fulfilled.

\* 4/1-2 Never leave titles/subtitles without text. Don't use uncommon acronyms for a title. All acronyms need to be previously defined in the text.

RESPONSE: Done. We changed the title to "2.1 A mobile measurement system for carbon dioxide". We deleted the subheading "2.1.1 System requirements" which now becomes the introductory text for 2.1.

\* 4/4 This sentence is repetitive.

RESPONSE: Due to the shortening of the section on previous work (Comment on 3/3 – 3/17) this sentence is no longer repetitive.

\* 4/5 What about "passenger" instead of "various"?

- RESPONSE: Done. Changed "various cars" to "passenger cars"

\* 4/7 Define limited time-scale and fine resolution. One, two five hours? Grid cells of 50, 100, 500 m2?

RESPONSE: Done. We do not specify a typical temporal range (deleted '(hours)')

because this depends on the size of the area to be measured. But given the density of our transect and typical vehicle speed, our 12.7 km2 area was sampled within 3.5 hours with five sensors. This means for one km2 and one sensor we can cover approximately 0.7 km2 hr-1. We added the following sentence: 'With typical vehicle speed and a characteristic urban street layout / traffic density, one sensor is capable of covering between 0.5 and 1 $\rm{km}^2$ per hour.'

* 4/9-10 ... have been stationary or mounted in specialized vehicles. . .

- RESPONSE: Agree. Changed as proposed.

* 4/13-16. Rewrite the whole paragraph. For example: Carbon dioxide analysers based on infrared detection (Licor...) were coupled with microcontrollers (Arduiono ...), etc.

RESPONSE: Done. The paragraph was rewritten in passive voice and simplified as follows: "We used a commercially available carbon dioxide infrared gas analyser (IRGA) (Li-820, Licor Inc., Lincoln, NE, USA). The Li-820 is a compact (23.23 x 15.25 x 7.62 cm, 1 kg), low maintenance (approx. 2 years of continuous use) and high accuracy (+- 1 ppm) CO2 analyzer. The Li-820 uses a single path infrared light to determine the CO2 mixing ratio within a closed path by detecting the amount of absorption of the light from the path. The gas analyzer was coupled with an Arduino microcontroller (Arduino CC, Ivrea, Italy). The Arduino platform is capable of communicating digitally with the IRGA, a Global Positioning System (GPS) unit (Adafruit Ultimate GPS Logger Shield with GPS Module, Manhattan, New York, USA) unit, and a digital temperature thermometer (Maxim Integrated One Wire Digital Temperature Sensor - DS18B20, San Jose, CA, USA). A custom hardware board was developed to connect all of the components together to distribute the correct amount of power to each of the hardware components and to allow for compact hardware and sensor input. The portable CO2 system was named the "Do-It-Yourself-Sensor-CO2", or "DIYSCO2' system. . ."

* 4/14 12.23 $\times$ 15.25 $\times$ 7.62 cm

- RESPONSE: Done. Changed to "12.23 × 15.25 × 7.62 cm" as proposed.

\* 4/16 No need "...monitoring applications including agriculture."

- RESPONSE: Done. Removed: "built for various CO2 monitoring applications including agriculture "

\* 4/19 Define GPS.

- RESPONSE: Done. Changed to "...Global Positioning System (GPS) unit..."

\* 4/27 Material of the tube?

- RESPONSE: Done. Material defined as "Synflex, Polyethylene/Aluminum composite"

\* 5/Panel b. Which is the height of the sampling line over the vehicle's roof?

RESPONSE: Done. Added to text: 'The sampling line inlet was 70 cm over the vehicle's roof and 2.2 m above the road surface'.

\* 5/Figure 1 legend. It is clear that both panels show photos of the CO2 system. No need of indicating that during operation the system is enclosed in the case neither that the used vehicle was a car-shared one.

RESPONSE: Done. We removed the sentence "During operation, the system is enclosed in the case, while LEDs on the box indicate system state" and also removed "of a car-sharing vehicle" and replaced by "of the vehicle".

\* 6/10 Indicate that in Canada, the driver position is at the left side.

RESPONSE: Done. We added a bracket "passenger window (right side)"

\* 6/19-26 Rewrite the whole paragraph.

RESPONSE: Done. Changed and simplified to "The study area is a 12.7 km x 1 km quadrangle within the City of Vancouver, BC, which spans from the northern-most tip of the city in forested "Stanley Park" (49° 18' 45.17"N, 123° 09' 29.10" W, WGS-84) to

the city's south eastern neigborhood "Victoria - Fraserview" (49°,12' 59.00"N, 123° 03' 46.90"W) (Fig. ... "

\* 6/20 Provide locations using latitude and longitude in degrees.

RESPONSE: Done. All coordinates in the entire text of the manuscript have been changed to degrees (WGS-84). The geodetic datum (WGS-84) is only defined in the first instance.

\* 6/27 Flux tower measurements.

RESPONSE: Done. Change title to "Flux tower measurements" as proposed.

\* 6/28 Eddy covariance flux tower . . .

RESPONSE: Done. Changed from "eddy-covariance tower" to "eddy covariance flux tower".

\* 6/28 No need to include the Fluxnet ID of the flux tower. The manuscript never makes reference to Fluxnet.

RESPONSE: Disagree. The reference given refers to the Fluxnet database. We kept the Fluxnet ID, because the Fluxnet ID is an official, unique global code, issued by ORNL that identifies the site. Knowing the Fluxnet ID goes along with the global fluxnet database that also hosts all data from this site, see: https://fluxnet.ornl.gov/site/4132.

\* 6/28 – 8/9 This paragraph needs to be rewritten. Indicate since then the flux tower has been working. Provide a reference to a comprehensive description of the tower.

RESPONSE: The reference to Crawford and Christen, 2015 is appropriate to describe the tower location. The paragraph has been rewritten as follows: "On the flux tower, a CSAT-3 ultrasonic anemometer-thermometer (Campbell Scientific Inc., Logan, UT, USA) provided continuous measurements of sensible heat flux (H), wind direction, and wind velocity. Further, air temperature (T_tower) was measured with a shielded HMP 45 thermometer / hygrometer (Vaisala Inc., Vanta, Finland). All four radiation components, including long-wave upwelling radiation (L_downarrow), were measured by a CNR-1 net radiometer (Kipp & Zonen, Delft, The Netherlands). Carbon dioxide molar mixing ratios r_tower were measured near tower top (28 m) using a tube that pumps air to a TGA200 closed path analyzer (Campbell Scientific Inc.). In addition, $CO_2$ mixing ratios were measured by a Licor-7500 open path IRGA (Licor Inc., Lincoln, NE, USA) co-located with the ultrasonic anemometer-thermometer."

* 7/Figure 2 legend. Avoid repeating information already given in the text e.g., 12.7 $\times$ 1 km). Try to use active sentences as much as possible (e.g., The location of the flux tower is marked by ... instead of Shown are also ... ).

RESPONSE: Done. Changed to active sentences. Deleted redundant information, i.e. "a 12.7 km x 1 km" and "(where all five systems were cross-checked before and after the campaign)".

* 7/Figure 2 legend. Crawford and Christen, 2014 is a very good paper, but I do not consider necessary to include it here.

RESPONSE: Done. Removed "and 24 hour measurements of $CO_2$ storage by Crawford and Christen (2014)."

* 8/11-15 Rewrite the whole paragraph. For example: Two fields campaigns took place, the first on ... during the summer and when trees leaves are in full, while the second on ... , covering the heating season. Sampling was conducted from 10:00 – 13:30 h, when vehicular traffic and meteorological conditions are relatively constant.

RESPONSE: Done as proposed, except that we kept the brackets to retain information on heating and leaf state: "Two field campaigns took place, the first on 28 May 2015 (non-heating season, broadleaf vegetation with leaves emerged) and the second on 18 March 2016 (heating season, before leaf emergence). For simplicity, data sets from the two dates will be referred to as "summer" (28 May 2015) and "winter" (18 March 2016). Sampling was conducted from 10:00 - 13:30 h, when vehicular traffic

and meteorological conditions are relatively constant."

* 8/16. Remove this sentence.

RESPONSE: Done. Removed "In order to ensure that the study area was comprehensively sampled during the duration of the measurement campaign"

* 8/17 Replace predefined by designed.

RESPONSE: Done. Changed to "designed"

* 8/16-19 Rewrite these two sentences.

RESPONSE: Done. Rearranged the paragraph as follows: "Five DIYSCO2 systems were installed on vehicles. Each of the five vehicle was assigned a route to travel approximately 70 km during the study period (achieving an optimal sampling density of about 3.5 km2 hr-1). Each vehicle started and ended at the southeast corner of the transect (49° 13' 15.08" N, 123° 04' 14.11"W, Fig. 2). The routes of the five systems were drawn such that a majority of the streets and lanes in the study area would be sampled at least once in the 3.5 hour time period, but ideally sampled at different times throughout the campaign. The routes were evaluated using an overlaid 100 m $\times$ 100 m grid, confirming that nearly all of the grid cells would be crossed by at least one system if the routes were successfully completed. Furthermore, a bicycle was used to traverse trails in the forested area of "Stanley Park" to sample along pathways in the densely forested ecosystem away from roads. "

* 8/23 Should be bicycle instead of bike?

RESPONSE: Done. Changed all instances of "bike" to "bicycle" in entire manuscript.

* 9/3 Describe briefly such filtering methods.

RESPONSE: Done. They are described in the following sentences, but this was unclear. Updated text reads as follows: "The 1 Hz-data from all five DIYSCO2 systems were filtered according to Crawford and Christen (2014), so that all data were removed

when the GPS recorded speeds were below . . ."

* 9/20 Data from the eddy covariance tower are. . .

RESPONSE: Done. Changed to " Data from the eddy covariance tower are"

* 10/Eq. 3 So many conversion factors at the beginning of the equation are * confusing. Merge them in only one factor and explain its meaning in the text.

RESPONSE: Disagree. We argue that readers benefit from knowing how the factor is calculated, which is only possible when separating the individual conversions.

* 10/15 Remove "in the two measurement campaigns". It is obvious.

RESPONSE: Done. Removed.

* 10/26-30. Make shorter this sentence. For example: This assumption is supported by a previous study in which no storage flux was observed during daytime for this particular site (Crawford and Christen, 2014).

RESPONSE: Done. We changed the sentence as suggested. The following sentence was also rewritten as "However, this assumption is severely violated at night and in the early to mid morning (Crawford and Christen, 2014; Bjorkegren et al., 2015), so the proposed approach does only work midday or afternoon."

* 11/1-3 Any reference on the Reynolds analogy?

RESPONSE: Done. We added a references to Arya's textbook (2001) and specify that this is the Reynolds analogy between turbulent heat transfer and passive scalar transfer.

* 11/11 Better indicate the percent of excluded or included readings.

RESPONSE: Done. Included this information as follows: "Grid cells with less than 10 samples were removed from further analysis, which resulted in 30.8% of all cells being removed in the summer campaign and 27.4% in the winter campaign." * 14/Table 2.

Indicate which those climate zones are.

RESPONSE: Done. Instead of numbers we now use the LCZ names according to Table 2 in Stewart and Oke (2012).

* 23/17 ... between 9:00 and 19:00 h ... This might be true for office buildings, but not for residential buildings.

RESPONSE: Agreed. We changed the sentence as follows to be more precise: "... studies show that building occupancy (and therefore energy use) varies. For example for office buildings, the major activity is between 9:00 and 19:00 (Martani et al., 2012)"

* 23/30 ... the traffic count data do not indicate ...

RESPONSE: Done. Changed "does" to "do"

* 24/7-9 Do not list what others have made (you did it already in the introduction), better say that this study was the first in combining ... . to evaluate CO2 emissions at fine scale.

RESPONSE: Done. We changed the text as follows: 'In this study, we proposed and implemented a new approach to determine and map CO2 emissions at fine scale across a city. The approach combines multiple mobile sensors at street level with an eddy covariance flux tower." Also the following sentence was adjusted accordingly: "A portable, mobile sensor system to measure the spatial variability of CO2 mixing ratios called the DIYSCO2 was developed and tested."

* 25/19 Replace six tanks by six standard gases

RESPONSE: Done. Changed from "six tanks" to "six standard gases"

* 27/12 ...to the Open Street Map (OSM)...

RESPONSE: Done. Added acronym in bracket "(OSM)"

**Reviewer 2: Anonymous**

Review of Lee at al. "A mobile sensor network to map CO2 emissions in urban environments" The paper presents a detailed methodology for measuring spatial CO2 emission in an urban landscape using low-cost sensor system deployed on vehicles in urban areas. Methodology for the estimation of CO2 fluxes from urban areas is compared with EC approaches measured using a traditional flux tower over Vancouver. This is a very detailed description of the approach and the validation against established methods and one can see the extreme usability of such a system. The paper is well within the remit of amt and the authors have a novel approach. This type of work is needed to address the major challenges of addressing the study of the urban atmosphere, not least its spatial heterogeneity. The authors discuss the various advantages and disadvantages of their approach. I am very happy at the level detail shown by the authors especially with the design and construction of the DIYSCO2 system. I would suggest the authors put the appendix in the supplementary information section of the article.

RESPONSE: We are thankful and appreciative of the comments provided. We recognize the feedback to move the appendix to the supplementary materials however given the feedback of the community (see comment by L. Golston below) and to emphasize the nature of AMT on measurement systems, we would like to keep the Appendix in the main, peer-reviewed part of the text.

**Community comment by L. Golston**

I would like to add some comments to help strengthen the description of the DIYSCO2 system:

* Appendix A1 is titled 'sensor precision', the first paragraph says accuracy is ensured by calibration but precision and linearity need to be tested, then the second paragraph says accuracy and linearity are tested, and the final paragraph concludes about linearity – please sort out the usage of each of these terms.

RESPONSE: Agreed. Corrected. In this section we have changed the title of the

section to, "A1: Sensor accuracy and linearity" and changed the text to, "The accuracy of the Li-820 is ensured using a two-point calibration, performed in the lab using a zero-gas and a standard span gas in the range of assumed measurement. In the current study, all standard tanks have been calibrated against primary CDML / NOAA WMO traceable tanks with a typical error between standard and primary tanks in r < 0.15 ppm." The observed data shown below in Table R2 indicates a strong linearity (R2 of 0.9999) and a root mean square error (RMSE) of 0.233 ppm for the six different CO2 mixing ratios. Note that sensor drift of the system are described in the following section, now renamed to "Sensor drift".

* 6/5 and Appendix A1: This at first glance suggests impressive linearity, however there is not enough information to evaluate this claim. As an extreme example, if five of the six tanks had a mixing ratio of 399.08 ppm and the sixth was at 503.77 ppm you could get a linear R2 of 0.9999 even with a nonlinear sensor - please state the values of each tank. It would also be helpful to know how the standard tanks were calibrated against CDML / NOAA traceable tanks as was stated:

RESPONSE: Agree. Done. It is important to clarify which tanks we used. that although we used 6 tanks, only four were different. The accuracy of the CDML/NOAA tanks were tested against the TGA200. The table R2 shows each of the tank's mixing ratios, its uncertainty, and the observed values from one of the Li-820. The text has been changed as follows. To test the linearity in the range 400 to 500 ppm, a test was performed using six standard gases of known $r$ at 400 (2 tanks), 413 (1 tank), 457 (2 tanks) and 504 ppm (1 tank).

* Appendix A3 states a time response of 2.2 seconds for a 50% step change within the IRGA. This indicates that the effective number of samples is less than the 1 Hz that is stated, and could have an implication for the effective sampling density of measurements in the city. Please include a statement about whether the time response of the sensor being > 1 s affects the results.

[Figure]

RESPONSE: Done. The 50% response time of the IRGA to a step change is 2.2 seconds, so the time constant is 3.2 s. So effectively, the resolution is less than 1 s as correctly pointed out. We changed the description in the main text and wrote a "nominal sampling rate" of 1 Hz and added a note that the sensor's physical time constant is however 3.2 s. Further in APpendix A2 we stated the positional error, assuming the typical vehicle speed "However, as the time constant with 3.2 s was higher than the nominal sampling frequency of 1 Hz, the actual sampling frequency was less than one second, leading to a positional standard deviation of the signal of 10 m, not 5 m (at typical speed of 20 km $\rm{hr}^{-1}$)." Note that there was also an overall delay of the system (which includes the 3m long sample inlet tube) of 18.2 seconds. We correct our measurements during post-processing from data gathered from field campaigns in order to shift the GPS and the timestamps of observed CO2 mixing ratios to properly attribute the measurements spatially and temporally.

* Appendix A4 This is an interesting test, but the second paragraph is somewhat difficult to read and several things need to be clarified: What exactly is the difference between the tests in p26/L30 where good agreement was seen between the grouped inlets, and p27/L1 where the grouped inlets had relatively large spread? By higher CO2 mixing ratios, does this mean this second results were for an area with direct traffic emissions?

RESPONSE: Done. Your understanding is correct. In areas without direct traffic emissions, there is good agreement between the sensors (+/- 0.5ppm), but in areas with direct traffic emissions, the varibility increases to the percentages listed. We rephrased accordingly. We also added a concluding statement "In summary, the sampling location is a source of much greater uncertainty than instrument accuracy, drift, or linearity in the context of this work."

* What about the results of the ungrouped test?

RESPONSE: Done. We added the following information "The results of the ungrouped inlet test showed that 54.98%, 79.08%, and 87.49% of the data have a variability within

5, 15, and 25 ppm, respectively for the data collected at 1s. When aggregated to 1 min, the data showed 66.67%, 91.66%, and 94.44% of the data are within 5, 15, and 25 ppm, respectively."

* p27/L1 implies that slightly less than half of the 1-s data is within 5 ppm, contradicting the next sentence. I also disagree with the use of 'accuracy' and 'error' here, since there is no standard to compare against other than the mean value, and would recommend spread or variability instead.

RESPONSE: Thank you. Noted. We have corrected the wording to "This indicates that slightly less than half of the 1-s data measured by the sensors are within 5 ppm of each other and that we can expect a majority of the data (>88.85%) to have variabilities up to 15 ppm depending on where on the car the inlet is mounted.

* If the authors move the appendices to a supplement per reviewer 2's comment, I would suggest retaining Appendix A since this would give a disproportionately large amount of space to the field results and comparison to inventory, in comparison to description of DIYSCO2 and the emission calculation methodology.

RESPONSE: We agree and would also ensure an equal treatment of sensor system design and testing and subsequent validation. Also we prefer to retain all appendices to keep them in the peer-reviewed part of the contribution.

* 2/15: This says accuracy, but the value given in parentheses is the 1-s noise specified in the instrument datasheet, not its accuracy. The Li-820 manual gives accuracy specifications, based on mixing ratio range and cell pressure, as a percentage (%) of reading

RESPONSE: Done. Corrected. Thank you for flagging this. It has been corrected in the text.

* 8/29: "In-situ calibration". This is a good thing to do, but should be called a comparison not a calibration since there is no standard used to calibrate against. Can also a

comment also be made about whether this provided useful information? If this means parking five vehicles adjacent to one another, and given the variability described in

RESPONSE: Done. Well appreciated. Changed "calibration" to "comparison". During the lab tests, the maximum drift over a 3.5 hour period was 0.82 ppm. We added the following text "During the field experiments however, we observed a maximum drift of +0.95 ppm relative to the mean of all sensors, which was greater than what was found in the lab test."

* Appendix A4, I imagine it might be difficult to detect drifts this way both for individual sensors and for the ensemble of sensors.

RESPONSE: True, it is difficult to assess drifts for individual and ensemble sensors, especially in the field and could be an interesting future approach to pursue. In this study, our method was to to compare the the individual measurements against the mean. No absolute reference was available other than the manual calibration against standard tanks.

* 12/5: What is the meaning of "sample" in this section. Does one sample correspond to one 1 Hz measurement? If so, there should be some discussion about vehicle speed since that will affect the spacing of measurements.

RESPONSE: Done. Correct, the sample corresponds to one 1hz measurement. We added a claifying bracket: "(1 sample equals one 1 Hz measurement)". We discuss the delay time and the method for correctly attributing the measurements in our discussion of the response time of the sensor system in Appendix A3. Here we added "At the average vehicle speed of 20 km hr-1 this corresponds to a spatial spacing of 5.5 m."

* Spell out IRGA at first use

RESPONSE: Done. IRGA is defined as "infrared gas analyzer".

* p2/L14 and p8/L5: NB -> NE

RESPONSE: Done. Changed to "NE" for Nebraska.

\* p9/L2: gridding

RESPONSE: Done. Changed all instances of "griding" to "gridding".

Please also note the supplement to this comment:
http://www.atmos-meas-tech-discuss.net/amt-2016-200/amt-2016-200-AC1-supplement.pdf
* * *
**Table R1 -** Student's t-test for pairs of neighborhoods (on 100 x 100 m grid cells). W = populations are statistically significantly different in winter; S = populations are statistically significantly different in summer campaign.

|  | SP | WE | DT | FM | KR | SV |
|---|---|---|---|---|---|---|
| Stanley Park (**SP**) |  | W | W / S | W / S |  |  |
| West End (**WE**) | W |  | W / S | S | W / S | W |
| Downtown (**DT**) | W / S | W / S |  | W / S | W / S | W / S |
| Fairview - Mount Pleasant (**FM**) | W / S | S | W / S |  | W / S | W / S |
| Kensington-C. Cot. - Riley Park (**KR**) |  | W / S | W / S | W / S |  |  |
| Sunset - Victoria-Fraserview (**SV**) |  |  | W / S | W / S |  |  |

**Fig. 1.** Table R1

**Table R2 -** Comparison between standard tanks and Li-820 observed CO2 mixing ratios.

| Tank ID | Standard tank mixing ratio (ppm) | Tank uncertainty relative to primary reference (ppm) | Li-820 observed (ppm) |
|---|---|---|---|
| UBC CO2-007 | 399.079 | ±0.047 | 400.38 |
| UBC CO2-010 | 400.340 | ±0.042 | 401.84 |
| UBC BIOMET-73 | 412.714 | ±0.112 | 413.99 |
| UBC CO2-011 | 456.912 | ±0.107 | 458.38 |
| UBC CO2-008 | 457.756 | ±0.131 | 458.67 |
| UBC CO2-001 | 503.767 | ±0.025 | 504.13 |

**Fig. 2.** Table R2

**Supplement:**

**A mobile sensor network to map carbon dioxide emissions in urban environments**

Joseph K. Lee[1], Andreas Christen[1], Rick Ketler[1], and Zoran Nesic[1,2]

[1]Department of Geography / Atmospheric Science Program, The University of British Columbia, Vancouver, BC, Canada
[2]Biometeorology Group, Faculty of Land and Food Systems, The University of British Columbia, Vancouver, BC, Canada

*Correspondence to:* A. Christen (andreas.christen@ubc.ca)

**Abstract.** A method for directly measuring carbon dioxide ($CO_2$) emissions using a mobile sensor network in cities at fine spatial resolution was developed and tested. First, a compact, mobile system was built using an infrared gas analyzer combined with open-source hardware to control, georeference and log measurements of $CO_2$ mixing ratios on vehicles (car, bicycles). Second, two measurement campaigns, one in summer and one in winter (heating-season) were carried out. Five mobile sensors were deployed within a $1 \times 12.7\,\mathrm{km}$ transect across the City of Vancouver, BC, Canada. The sensors were operated for 3.5 hours on pre-defined routes to map $CO_2$ mixing ratios at street level, which  were then averaged to $100 \times 100$ m  grid cells. The averaged $CO_2$ mixing ratios of all grids in the study area were 417.9 ppm in summer and 442.5 ppm in winter. In both campaigns, mixing ratios were highest in the grid cells of the downtown core and along arterial roads and lowest in parks and well vegetated residential areas. Third, an aerodynamic resistance approach to calculating emissions was used to derive $CO_2$ emissions from the gridded $CO_2$ mixing ratio measurements in conjunction with mixing ratios and fluxes collected from a 28-m tall eddy-covariance tower located within the study area. These  measured emissions showed a range of -12 to 226 $\mathrm{kg\,CO_2\,ha^{-1}\,hr^{-1}}$ in summer and of -14 to 163 $\mathrm{kg\,CO_2\,ha^{-1}\,hr^{-1}}$ in winter, with an average of 35.1 $\mathrm{kg\,CO_2\,ha^{-1}\,hr^{-1}}$ (summer) and 25.9 $\mathrm{kg\,CO_2\,ha^{-1}\,hr^{-1}}$ (winter). Fourth, an independent emissions inventory was developed for the study area using buildings energy simulations from a previous study and routinely available traffic counts. The emissions inventory for the same area averaged to 22.06 $\mathrm{kg\,CO_2\,ha^{-1}\,hr^{-1}}$ (summer) and 28.76 $\mathrm{kg\,CO_2\,ha^{-1}\,hr^{-1}}$ (winter) and was used to compare against the measured emissions from the mobile sensor network. The comparison on a grid-by-grid basis showed linearity between $CO_2$ mixing ratios and the emissions inventory ($R^2 = 0.53$ in summer and $R^2 = 0.47$ in winter).  Eighty seven percent (summer) and 94% (winter) of measured grid cells show a difference within $\pm$ 1 order of magnitude, and 49% (summer) and 69% (winter) show an error of less than a factor 2. Although associated with considerable errors at the individual grid cell level, the study demonstrates a promising method of using a network of mobile sensors and an aerodynamic resistance approach to rapidly map greenhouse gases at high spatial resolution across cities. The method could be improved by longer measurements and a refined calculation of the aerodynamic resistance.

**1 Introduction**

Cities and the cumulative processes of urbanization are key drivers of local and global environmental change (Mills, 2007; Grimmond, 2007). As cities are the centers of increasing population growth and resource consumption, they are also the dominant source of greenhouse gas emissions - in particular carbon dioxide ($CO_2$) - into the atmosphere (Rosenzweig et al.,

5   2010). On the global scale, urban  areas are responsible for up to 80%  of the total anthropogenic $CO_2$ emissions footprint (Satterthwaite, 2008). Cities are thus responsible for a major proportion of the anthropogenic greenhouse gas emissions that are intensifying positive atmospheric radiative forcing of the troposphere contributing to global climate change (IPCC, 2013), although a large fraction of the emissions related to the resource chains that sustain cities does not occur within the built-up area, but rather is emitted elsewhere.

10    Within cities, the major sources of $CO_2$ are the combustion of fossil fuels for heating, ventilation, and cooling systems (HVAC), transportation, industrial processes, and power generation (Kennedy et al., 2009).  These fossil fuel emissions are combined with $CO_2$ emitted from biological sources, namely soil, plant and human respiration and in part taken up by photosynthesis of urban vegetation (Christen et al., 2011). Overall, fossil fuel sources dominate  $CO_2$ fluxes in cities. The sequestration of $CO_2$ by urban vegetation in most cities is very limited (Velasco et al., 2016). However, the rate of

15   $CO_2$ uptake by photosynthesis  at a given time, can be relevant and is measurable in highly vegetated cities during the daytime in the growing season (Peters and McFadden, 2012; Weissert et al., 2014). The dominance of fuel emissions results in increased concentrations of $CO_2$ in the urban boundary layer (UBL) relative to rural or pristine air (Idso et al., 2001; Grimmond et al., 2002; Vogt et al., 2006). The enrichment of $CO_2$ in the UBL links directly to emissions which are controlled by urban form and function.

20   With more than 50% of the global population now living in cities (United Nations, 2014), cities are also the place where effective mitigation of climate change, driven by policy, design, and bottom up citizen engagement is possible.  According to IPCC (2014), the urban scale has the highest potential for  fast, efficient, and sustained implementation of mitigation efforts. Central to the reduction of urban $CO_2$ emissions is the availability of reliable emissions information and inventories and methods of validating city-scale emissions estimates and reduction efforts. While there are a growing number

25   of methods of quantifying emissions in urban areas, there are disconnects between the current spatial and temporal resolution of emissions models, the ever-evolving urban form and function, and block to neighborhood-scale measurements which inform and validate emissions models (Pataki et al., 2009; Kellett et al., 2013). It further remains a challenge to directly measure emissions at fine urban scales and separate  $CO_2$ emission measurements in the urban atmosphere into different fossil fuel emissions and biological sources (Christen, 2014).

30   The  research goal of this  study is to develop, apply and test a new methodology to map $CO_2$ emissions in complex urban environments.  Data from a network of mobile sensors and  an eddy covariance flux tower combined with an aerodynamic approach are used to calculate and map emissions at fine scales (blocks to neighborhoods) in cities.

Mobile  measurements have been used in the past for studying and mapping the spatial variability of greenhouse gases in cities (Jimenez et al., 2000; Idso et al., 2001; Henninger and Kuttler, 2007; Crawford and Christen, 2014).  Because trace gas analyzer systems for greenhouse gases

5  are still bulky  (Tao et al., 2015, e.g. ), past mobile mapping studies utilized specialized research vehicles (Bukowiecki et al., 2002; Elen et al., 2013; Crawford and Christen, 2014). While these  vehicles have the advantage that they can be  equipped with additional components such as

10 calibration tanks or computers,  the complexity of such systems does not allow for easy deployment  on standard and flexible modes of transport.

There  is increasing interest to develop innovative methods for monitoring urban climate and air pollution using  low-cost distributed sensor networks.  For example,

15 ~~measures of rainfall for the entire Netherlands using the attenuation of a cell phone sender signal to its receiver station. In another example, Overeem et al. (2013) developed methodology to derive fine-scale air temperature measurements using cell phone battery temperatures to examine the urban heat island. Bottom-up approaches using distributed sensor networks have become possible in recent years with the increasing availability of low cost climate and air pollution sensors, open source programmable microcontrollers, and improvements in networking infrastructure. For example,~~ Meier et al. (2015) used sensor

20 data from a commercial consumer-grade weather station network to examine fine-scale urban heat island effects in the city of Berlin. In another example, Chapman et al. (2015) developed a road sensor network to monitor road surface temperatures to optimally salt roads during the winter months in Birmingham. Given this growing interest in distributed  sensing systems and the advances in  the related technologies, could there be new opportunities for the fine-scale mapping of CO$_2$ emissions in cities?

25 ~~enough to be integrated into existing infrastructure such as bikes, car-sharing cars, taxis, or even autonomous flying vehicles? Hence, the key considerations for developing new mobile CO$_2$ emission monitoring systems must be around scalability (how many can be built and for what cost?), system extendability (can the system be built upon?), accuracy and precision, temporal resolution, accessibility (e.g open source or proprietary?), and the mobile platform on which the sensor is to be mounted.~~

 This study investigates whether it is  feasible to map green-

30 house gas emissions, specifically CO$_2$, at a spatial resolution of neighborhoods / blocks across the city with a portable network of mobile sensors that  can be routinely implemented on  various mobile platforms. In order to address  this question, four major objectives were  pursued:

1. Sensor Development: Develop and test a compact, mobile, and multi-modal CO$_2$ sensor for  bicycles and cars.

2. Measurement Campaign: Deploy the sensors in a targeted measurement campaign.

3. Methodology development: Calculate emissions from measurements of $CO_2$ mixing ratios and aerodynamic resistance (in the following called "measured emissions")

4. Analysis and Evaluation: Compare the measured emissions to fine-scale traffic and building emissions inventories. Can we find agreement between the spatial patterns in the inventories and measured emissions?

**2 Methods**

**2.1 The  mobile measurement system for carbon dioxide**

**2.1.1**

A mobile $CO_2$ monitoring system was required to address the project's need for multiple, low cost, yet accurate sensors capable of measuring mixing ratios and position at high frequency (≈ 1 Hz to have an error of 5 m at typical driving speeds) and easily deployable on  bicycles and passenger cars with a compact design. A mobile monitoring system with such specifications is necessary to cover large geographic areas within limited time scales  at sufficiently fine resolution that are representative of typical urban emission patterns. With typical vehicle speed and a characteristic urban street layout / traffic density, one sensor is capable of covering between 0.5 and 1 $km^2$ per hour. Sensor systems with many of these specifications do already exist, but few, if any, were designed to be carried on and easily interface with various types of mobile platforms; all studies using high accuracy $CO_2$ sensors either have been stationary or  mounted in specialized vehicles because of the weight, power consumption, and size of the sensors being used and are highly costly.

**2.1.1 System design**

 We used a commercially available carbon dioxide infrared analyzer (IRGA) (Li-820, Licor Inc., Lincoln, NE, USA). The Li-820 is a compact (23.23  × 15.25  × 7.62 cm, 1 kg), low maintenance (approx. 2 years of continuous use) and  low noise (± 1 ppm)  $CO_2$  analyzer (Li-Cor, 2015). The Li-820 uses a single path infrared light to determine the $CO_2$ mixing ratio within a closed path by detecting the amount of absorption of the light from the path.  The Li-820 was operated with a nominal sampling rate (data output) of 1 Hz but the actual time constant of the system was determined 3.2 s (see Appendix A2). The gas analyzer was coupled with an Arduino microcontroller (Arduino CC, Ivrea, Italy). The Arduino platform is capable of communicating digitally with the IRGA, a  Global Positioning System (GPS) unit (Adafruit Ultimate GPS Logger Shield with GPS Module, Manhattan, New York, USA) unit, and  a digital temperature thermometer (Maxim Integrated One Wire Digital Temperature Sensor - DS18B20, San Jose, CA, USA). A cus-

[Figure]

[Figure]

**Figure 1.** (**a**) Photo of the "DIYSCO₂" system (case open) with components labelled.  (**b**) Inlet mounted through the passenger window (right side) of  the vehicle, the "DIYSCO₂" sits in the trunk space.

tom hardware board was developed to connect all of the components together  to distribute the correct amount of power to each of the hardware components  and to allow for compact hardware and sensor input. The portable $CO_2$  system was named the "Do-It-Yourself-Sensor-$CO_2$", or "DIYSCO₂" system (Fig. 1a)

5     The DIYSCO₂ system reports $CO_2$ as mixing ratios ($r$) in ppm, geoposition (latitude/longitude, speed, altitude, and satellite strength), and internal and external air temperature which are logged onto a micro-Secure Digital (SD) card at 1-second intervals. Air is drawn into the DIYSCO₂ system through a 3 m long inlet tube (6.35 mm diameter, Synflex, Polyethylene/Aluminum composite, Dekoron, Mt. Pleasant, TX, USA) using a small KNF NMP015 Micro-Diaphragm Pump (KNF Neuberger, Inc., Trenton, NJ, USA) first passing through a mesh filter at the sample inlet head

to prevent large particles from entering the DIYSCO$_2$ system (e.g. insects) and then through a Balston disposable filter unit (DFU) (Parker Hannifin Corporation, Lancaster, NY, USA) at the end of the 3 m tube. The flow rate is regulated by a Swagelok needle valve at 700 $\mathrm{cc\,min^{-1}}$ as recommended by Licor to minimize the effect of internal cell pressure changes on the CO$_2$ measurements. The entire DIYSCO$_2$ system is 35.8 cm x 27.8 cm x 11.8 cm, weighs 2.6 kg and is contained in a weather-proof

5 case (NANUK 910, Plasticase, Terrebonne, CA, USA). The system is powered by a single 9-18V DC/DC input which can be supplied by battery or via car cigarette lighter socket.

**2.1.2 System testing and installation**

Within the range of typical ambient mixing ratios of CO$_2$ between 400 and 550 ppm the DIYSCO$_2$ system showed strong linearity (R$^2$ of 0.9999) and a root mean square error (RMSE) of 0.233 ppm relative to six tanks of reference gases (see Ap-

10 pendix A1). The maximum sensor drift over three hours (the duration of the campaign, see below) under controlled conditions was in the range of -0.31 and +0.51 ppm (see Appendix A2). In the configuration used, the DIYSCO$_2$ had a time lag of 18.2 s between measurement intake and analysis (see Appendix A3).

Appendix A4 discusses errors associated with mounting the inlet at different positions on the car which can lead to a systematic bias. Generally, values on the driver side (centre of road) were higher than the passenger side. In the current work,

15 the sample inlet tube was run out through the passenger side window of the  vehicle. The sampling line inlet was 70 cm over the vehicle's roof and 2.2 m above the road surface (Fig. 1b). In order to deploy the DIYSCO$_2$ on a bicycle, the setup requires a 40 $\ell$ backpack to carry the sensor and a 7 Amp-hour, 12V gel-cell battery and a 1.5 m long rigid mounting tube (6 mm diameter) to mount the inlet tube above the cyclist. The sensor is placed in the backpack with the battery and worn on the back of the cyclist to reduce vibrations to the sensor system.

20 ## 2.2 Measurement campaigns

The systems were tested in two field campaigns. In each of the campaigns, a fleet of five sensors were operated simultaneously on pre-defined routes to evaluate the potential to map emissions and compare them against inventory data.

**2.2.1 Study area**

The study area  is a 12.7 km $\times$ 1 km quadrangle  within the City of Vancouver, BC,

25  which spans from the northern-most tip of the city  in forested "Stanley Park"  (49° 18′ 45.17″N, 123° 09′ 29.10″W, WGS-84) to the city's south eastern neigborhood  "Victoria - Fraserview" (. 49° 12′ 59.00″N, 123° 03′ 46.90″W) (Fig. 2). It includes dominant urban land uses - the downtown core, medium density residential, single detached residential, light industrial development, parks and forest. The study area  encompasses approximately 11.1% of the total area of the City of Vancouver,

30 and was selected, because of the provision of high resolution geospatial data, including LIDAR measurements of urban form

[Figure]

**Figure 2.** Map of the study area  (thick black outline).  Thin black lines refer to the paths of each of the five $DIYSCO_2$ systems. The colored areas are the neighborhoods used in further analysis.  The location of the eddy covariance tower and the start and end point of all paths are labelled. The $1.9 \times 1.9$ km box labelled "Sunset study area" refers to the domain of previous research, including the fine scale emission inventory developed by Christen et al. (2011) and Kellett et al. (2013).

used for building emission simulations in previous research (van der Laan, 2011), the availability of detailed traffic counts, and the location of a 30-m tall eddy-covariance tower .

**2.2.2  Flux tower measurements**

The  eddy covariance flux tower "Vancouver-Sunset",(ID: Ca-VSu FLUXNET (2016); Crawford and Christen (2015)) is located  near the south east corner of the study area (49° 13′ 34.0″N 123°04′42.2″W). On the flux tower,  a CSAT-3 ultrasonic  anemometer-thermometer (Campbell Scientific Inc., Logan, UT, USA)  measured continuously sensible heat flux ($H$), wind direction, and wind velocity. Further, air temperature ($T_{\text{tower}}$) was measured with a shielded HMP 45 thermometer / hygrometer (Vaisala Inc., Vanta, Finland). All four radiation components, including long-wave upwelling radiation ($L_\uparrow$), were measured by a CNR-1 net radiometer (Kipp & Zonen, Delft, The Netherlands). Carbon dioxide molar mixing  ratios $r_{\text{tower}}$  were measured near tower top (28 m) using a tube that pumps air to a TGA200 closed path analyzer (Campbell Scientific Inc.). In addition, $CO_2$ mixing ratios were measured by a Licor-7500 open path IRGA (Licor Inc., Lincoln, NE, USA) co-located with the ultrasonic anemometer-thermometer. The TGA200 was calibrated every 10 minutes against three WMO-traceable tanks of known $CO_2$ mixing ratios to ensure an accuracy of < 0.15 ppm. The Licor-7500 is calibrated twice a year in the lab. Further details of the site location, instrument exposure and data processing are discussed in Crawford and Christen (2015). Measurements on the flux tower made it possible to link mobile measurements with data from above the city and determine aerodynamic resistances for the calculation of emissions (see Section 2.4.1)

**2.2.3 Mobile measurements**

Two field campaigns took place,  the first on 28 May 2015 (non-heating season, broadleaf vegetation with leaves emerged) and  the second on 18 March 2016 (heating season, before leaf emergence). For simplicity, data sets from the two dates will be referred to as "summer" (28 May 2015) and "winter" (18 March 2016).  Sampling was conducted from 10:00  – 13:30  h, when vehicular traffic and meteorological conditions are relatively constant.

 Five DIYSCO$_2$ systems were installed on vehicles. Each of the five vehicles was assigned a route to travel approximately 70 km during the study period (achieving an optimal sampling density of about $3.5 \text{ km}^2 \text{ hr}^{-1}$). Each vehicle started and ended at the southeast corner of the transect (49° 13′ 15.08″ N, 123° 04′ 14.11″ W, Fig. 2).  The routes of the five systems were drawn such that  a majority of the streets and lanes in the study area would be sampled at least once in the 3.5 hour time period, but ideally sampled at different

times throughout the campaign. The routes were evaluated using an overlaid 100  × 100 m grid, confirming that nearly all of the grid cells would be  crossed by at least one system if the routes were successfully completed.

5  Furthermore, a  bicycle was used to traverse trails in the forested area of "Stanley Park" to sample along pathways in the densely forested ecosystem away from roads.

[revised manuscript text omitted]

In order to convert the molar flux $\overline{w'c'}$ (in $\mu\mathrm{mol\,m^{-2}\,s^{-1}}$) to a mass flux $F_c$ consistent with inventories (in $\mathrm{kg\,CO_2\,ha^{-1}\,hr^{-1}}$), we rewrite:

$$F_c = -M_c\, b_a\, b_t\, b_o\, b_m\, \frac{c_{\mathrm{tower}} - c_{\mathrm{mobile}}}{r_{aH}} \tag{3}$$

where $M_c$ is the molar mass of $CO_2$ ($44.01\,\mathrm{g\,mol^{-1}}$), $b_a$ is a factor for converting $\mathrm{m^{-2}}$ to $\mathrm{ha^{-1}}$ (i.e. $b_a = 10^4\,\mathrm{m^2\,ha^{-1}}$), $b_t$ is a factor for converting $\mathrm{s^{-1}}$ to $\mathrm{hr^{-1}}$ (i.e. $b_t = 3600\,\mathrm{s\,hr^{-1}}$), $b_o$ is the factor for converting $\mu\mathrm{mol}$ to mol (i.e. $b_m = 10^{-6}\,\mu\mathrm{mol\,\mu mol^{-1}}$) and $b_m$ is the factor for converting g to kg (i.e. $b_m = 10^{-3}\,\mathrm{kg\,g^{-1}}$).

[revised manuscript text omitted]

 Two percent and 16% of the measured $r_{mobile}$ were lower than the tower ($r_{tower}$) during the summer and winter campaign, respectively.  Three percent and 7% were higher than 500 ppm in summer and winter, respectively.

**3.1.2 Grid sample counts**

For the 100  × 100 m grid cells that could be traversed, in summer 91.31% of the grid cells contained more than 10 samples per grid cell (1 sample equals one 1 Hz measurement), 69.24% of cells contained more than 20 samples, and 28.32% of cell contained more than 50 samples. At the average vehicle speed of 20 km hr$^{-1}$, this corresponds to a typical spatial spacing of 5.5 m. For the winter campaign, 90.85% of the grid cells contained more than 10 samples, 72.64% contained more than 20 samples, and 27.36% contained more than 50 samples. Grid cells with less than 10 samples were removed from further analysis, which resulted in 30.8% of all cells being removed in the summer campaign and 27.4% in the winter campaign. Generally, grid cells along major roads tended to have more sample counts because they were traversed at different times, often by different vehicles.

**3.1.3 Grid averaged statistics**

Of the 1332 grid cells that could be traversed by a car or  bicycle, the case study covered 1024 in summer and 1037 in winter, of which 821 and 856 were further used (based on the condition of more than 10 samples). The maps of gridded $r_{mobile}$ for the summer and winter campaign are shown in Fig. 5. Table 2 summarizes the measured mixing ratios separated by neighborhood. In summer, the grid averaged $r_{mobile}$ of all valid  grid cells in the entire transect ranged between 393.1 ppm and 518.0 ppm, averaged 417.9 ppm, and had a median of 410.0 ppm. In winter, the grid averaged $r_{mobile}$ ranged between 408.4 ppm and 560.5 ppm, averaged 442.5 ppm.  Three percent of all grid cells in summer, and 8% in winter were showing a $r_{mobile}$ that was lower than $r_{tower}$, the majority of those cases were located in the forested "Stanley Park" in both campaigns (Tab. 2). Selected cells in the residential parts of "Riley Park / Kensington - Cedar Cottage" neighborhood were also showing a $r_{mobile}$ that was lower than $r_{tower}$.

Both campaigns showed considerable variation of $r_{mobile}$ between grid cells in the same neighborhoods. Overall, the grid cells covering major arterial roads and downtown core showed the highest maximum, minimum, median and mean $r_{mobile}$. Conversely, the grid cells covering residential streets and forested trails exhibited the lowest $r_{mobile}$ for the same statistics. Of all neighborhoods, "Kensington-Cedar Cottage / Riley Park" exhibited the lowest, and "Downtown" the highest average $r_{mobile}$ in both campaigns (Tab. 2).

Similarly, standard deviations within each 100 m grid cell (not shown) are highest along the major arterial roads and in "Downtown". In contrast, the residential areas have lower standard deviations within grid cells indicating less variability in $r_{mobile}$ for less busy roads. The trends are similar in the winter campaign except that there is overall higher standard deviation

**Table 2.** Grid-averaged mixing ratios ($r_{\text{mobile}}$), standard deviation of all grid cell means in the neighborhood, and fraction of cells with $r_{\text{mobile}} < r_{\text{tower}}$ per neighborhood

| Neighborhood | LCZ[a] | Mean mixing ratio $r_{\text{mobile}}$ (ppm) | | Std. dev. of $r_{\text{mobile}}$ (ppm) | | Fraction of cells with $r_{\text{mobile}} < r_{\text{tower}}$ | | Number of grid cells | |
|---|---|---|---|---|---|---|---|---|---|
| | | Summer | Winter | Summer | Winter | Summer | Winter | Summer | Winter |
| Stanley Park |  Dense trees | 413.7 | 435.6 | 19.1 | 24.3 | 4% | 28% | $N = 78$ | $N = 86$ |
| West End |  Compact high-rise | 416.1 | 442.7 | 15.1 | 15.9 | 1% | 4% | $N = 102$ | $N = 111$ |
| Downtown |  Compact high-rise | 437.8 | 474.9 | 19.2 | 26.5 | 0% | 0% | $N = 117$ | $N = 115$ |
| Fairview / Mount Pleasant large low-rise |  Open low-rise & | 421.2 | 446.2 | 19.0 | 17.6 | 0% | 0% | $N = 136$ | $N = 144$ |
| Kensington-C. C. / Riley Park |  Open low-rise | 411.0 | 432.3 | 13.5 | 15.1 | 1% | 11% | $N = 225$ | $N = 245$ |
| Sunset / Victoria-Fraserview |  Open low-rise | 413.3 | 434.7 | 14.2 | 16.0 | 0% | 8% | $
[revised manuscript text omitted]
  $1.9 \times 1.9$ km area, emissions were modelled  34.0 kg $CO_2$ ha$^{-1}$ hr$^{-1}$ and  measured emissions by eddy covariance were 30.8 kg $CO_2$ ha$^{-1}$ hr$^{-1}$ . The current study estimates emissions for the "Sunset / Victoria-Fraserview" neighborhood (that  is larger than the area in (Christen

et al., 2011) , Fig. 2) for March 18 (winter) as only 16.8 kg $CO_2$ $ha^{-1}$ $hr^{-1}$  . For the month of May, Christen et al. (2011) report modelled emissions of 26.9 kg $CO_2$ $ha^{-1}$ $hr^{-1}$ and measured emissions of 26.0 kg $CO_2$ $ha^{-1}$ $hr^{-1}$. The current study matches extremely well here, with emissions for "Sunset / Victoria-Fraserview" on May 28 (summer) of 26.5 kg $CO_2$ $ha^{-1}$ $hr^{-1}$. Note that  not only the spatial extent, but also 
[revised manuscript text omitted]

Data-driven models in combination with urban surface databases (urban form, traffic) could be used to further improve the information in the post processing and hence assist the derivation of more realistic emission maps (e.g. Moosavi et al., 2015).

**5 Conclusions**

   In this study, we proposed and implemented a new approach to determine and map $CO_2$ emissions at fine scale across a city. The approach combines multiple mobile sensors at street level with an eddy covariance flux tower.

A portable, mobile sensor system to measure the spatial variability of $CO_2$ mixing ratios called the $DIYSCO_2$ was developed  and tested. Five $DIYSCO_2$'s were deployed across a 12.7 $km^2$ study area over a period of 3.5 hours; the average sampling density was about 40 samples $ha^{-1}$. Of the 11.7 $km^2$ study area that could be traversed, 8.5 $km^2$ in summer and 8.2 $km^2$ in winter were sampled with $> 10$ samples per grid cell. Hence, excluding the grid cells with $< 10$ samples, the sampling density was roughly 0.5 $km^2 sensor^{-1} hr^{-1}$ over the 3.5 hour period for the 5 sensors. If it is assumed that this sampling density is appropriate for representing urban scale processes, it would require 230 coordinated mobile sensors on predefined routes to be deployed across the entire City of Vancouver (115 $km^2$) to measure $CO_2$ emissions across the city during the same time – obviously an effort that is not realistic.

However as sensor parts will become cheaper in the future, possibilities exist to integrate mobile sensor systems into operational vehicles such as taxis (e.g. 600 in the City of Vancouver) and mobility-on-demand services (e.g. currently there are >1000 carshare vehicles in the City of Vancouver). Alternatively, the time frame could be extended and using proper data selection, one could create composite maps from $r_{mobile}$ measured on different days under similar conditions. It would take 10 days in a coordinated effort to cover the entire City of Vancouver similar to the current transect.

A further question to be explored is whether the current number of samples ($> 10$ s) per grid cell is sufficient to represent the typical emissions in the cell given the intermittent traffic and the fact that large coherent structures are mostly responsible for mixing of pollutants out of the urban canopy layer (Salmond et al., 2005; Christen et al., 2007). Would a higher density of points (including multiple campaign days) improve the correlation between measured and inventory emissions?

The method to map emissions based on the aerodynamic resistance approach is sensitive to the measurements that are used to derive the aerodynamic resistance of heat and requires that a number of assumptions and conditions are met, yet, the work shows that the aerodynamic resistance approach can be used reasonably on a scale of $100 \times 100$ m grid cells to derive emissions from measures of aggregated mixing ratios. The measured emissions across the study area ranged from -12 $kg\,CO_2\,ha^{-1}\,hr^{-1}$ to 225 $kg\,CO_2\,ha^{-1}\,hr^{-1}$ per grid cell, thus showing the possibility for this methodology to detect negative emissions (net uptake), where photosynthesis is greater than the combined combustion and respiration emissions.

The research presented is proof of concept for a future in which atmospheric sensing is integrated into urban mobility. We haves shown the successful development of new technology and methodology for monitoring and mapping $CO_2$ mixing ratios and emissions in complex urban environments, at much finer scale than previously possible. Despite the simplicity of the methodology, the study demonstrated that it is possible to measure emissions across a complex landscape with a fleet of mobile sensors, an eddy-covariance tower, and the use of the aerodynamic approach to calculating emissions.

The data gained can be used to map and validate emissions as well as be integrated into regional efforts using observations and inversion modelling (Newman et al. (2013)) or even with total column measurements of $CO_2$ from satellites.

Further, the concept could be translated to the mapping of other trace gases and air pollutants emitted from vehicles and houses, air and surface temperature, and other environmental variables that affect human health, comfort, and safety. However, due to the assumption that sources are in the canopy layer where sensors operate, the proposed methodology is not necessarily transferable to emissions whose sources are not well represented such as fugitive natural gas emissions (methane) or volatile organic compounds or large industrial sources (tall stacks).

The development of smaller, more affordable, mobile sensor systems can facilitate new methodological approaches to monitoring the urban environment. With a fleet of mobile sensors and the methodologies for processing the derived datasets, the possibility to map and consequently validate emissions inventories is promising, as is the derivation or real time pollution and climate data in cities.

**Appendix A: Testing of sensor system**

Several key system specifications of the $DIYSCO_2$ were evaluated during the prototyping, namely: linearity, accuracy and drift, measurement lag time, between sampling and measurement, and the effects of inlet location on measurement variability.

**A1 Sensor accuracy and linearity**

The accuracy of the Li-820 is ensured using a two-point calibration, performed in the lab using a zero-gas and a standard span gas in the range of assumed measurement. In the current study, all standard tanks have been calibrated against primary CDML / NOAA WMO traceable tanks with a typical error between standard and primary tanks in $r$ of $< 0.15$ ppm. For the current application, accuracy and linearity of the Li-820 sensor is relevant in the range 400 to 500 ppm to enable comparisons between different $DIYSCO_2$'s operated simultaneously and also to properly compare $r_{\mathrm{mobile}} - r_{\mathrm{tower}}$.

To test the linearity in the range 400 to 500 ppm, a test was performed using six standard gases of known $r$ at 400 (2 tanks), 413 (1 tank), 457 (2 tanks) and 504 ppm (1 tank). All Li-820 sensors were first left running for 2 hours to account for the warm up time. The $DIYSCO_2$'s were connected to a calibration gas using a Union Tee connector. For each of the six gases, the calibration protocol called for an initial two minute system flush and then a recording of the values for at least 1 minute each. A minimum of 60 points per gas sample were used to calculate the average mixing ratios per tank measured by the system.

The Li-820 contained in the $DIYSCO_2$ showed strong linearity ($R^2$ of 0.9999) and a root mean square error (RMSE) of 0.233 ppm for the four different $r$. This indicates that the IRGA is operating well within its factory specifications of 1 ppm when calibrated and linearity  and accuracy are not the limiting factor for this type of study.

**A2 Sensor drift**

Sensor  drift is assessed to determine the $DIYSCO_2$'s ability to properly resolve the variability of mixing ratios during the duration of the campaign. Sensor drift was tested over the course of 7 days, with 5 sensors drawing in air from the same point outdoors at $\approx 3$ m in an urban context.

The RMSE between the five  systems at a 1-minute resolution ranged between 0.2 and 3 ppm for the seven day period and is therefore time dependent. The drift in the lab was up to $\pm 3.32$ ppm per day for individual sensors and days. Given that the field campaign was planned to be 3 to 3.5 hours long, the maximum drift of any sensor in any 3 hours was determined at most -0.31 ppm and $+0.51$ ppm relative to the mean of all 5 sensors. During the field experiments however, we observed a maximum drift of +0.95 ppm relative to the mean of all sensors, which was greater than what was found in the lab test.

**A3  Time constant and lag time**

The system measurement lag time is the time delay from when a measurement first enters the sample inlet of the system to when the signal is registered by the sensor. The $DIYSCO_2$'s measurement lag time is important to correctly attribute measurements to their geographic space.

For a given tube length and flow rate, the lag time will differ and therefore affect the system response characteristics. The values here are for a tube length of 3 m. Lab measurements were performed in which a solenoid switch was used to pass nitrogen gas with 0 ppm $CO_2$ into the sample tube inlet while simultaneously logging the exact second in which the solenoid was triggered. To calculate the lag time value for the system, the number of seconds were counted from when the sample enters the  tube until 50% of the change was reached.

The measurement lag time of the $DIYSCO_2$ system was determined to be 18.2 s. It took on average 16 seconds for the sample to travel from the inlet to the IRGA and a time constant $\tau$ of 3.2 s. We consequently used a value of 18 seconds in the post processing to shift the GPS and observed $r_{\mathrm{mobile}}$ time series to properly attribute measurements spatially to locations. However, as the time constant with 3.2 s was higher than the nominal sampling frequency of 1 Hz, the actual sampling frequency was less than one second, leading to a positional standard deviation of the signal of 10 m, not 5 m (at a typical speed of 20 km $\mathrm{hr}^{-1}$).

**A4  Effects of inlet location**

Two tests were performed to examine possible sampling biases due to different sample inlet locations on a vehicle. First, a test was done with five $DIYSCO_2$ in the same vehicle, where all the inlet tubes were bundled together at 2.2 m height, measuring at the same location  (within a few cm of each other) of the vehicle (referred to as "Grouped Inlet Test"). A second test was done  again with five $DIYSCO_2$ in the same vehicle, but with each of the inlet tubes located at different locations on the  vehicle (referred to as "Ungrouped Inlet Test"). Locations tested were all  again at 2.2 m height: One each above the driver's side front, driver's side back, passenger side front, and passenger side back window.

Both test were performed in the City of Vancouver using a Toyota Tacoma Truck along a route with traffic volumes ranging from 300 to 850 vehicles per hour.

In areas with a well-mixed atmosphere and on roads with little traffic, the $DIYSCO_2$ systems for the grouped inlet test showed a range within $\pm 0.5$ ppm of the mean all five sensors for 1 second data. For the ungrouped inlet test under those same conditions, the  range deteriorated to $\pm 5$ ppm of the mean.  Adding the higher traffic road segments, with observations of higher $CO_2$ mixing ratios, the standard deviation between all five of the $DIYSCO_2$ locations increases for the 1  Hz data. With inlets grouped together, 48.9%, 81.16%, and 90.14% of the one second data have  a standard deviation within 5, 15, and 25 ppm.  The results of the ungrouped inlet test showed that 54.98%, 79.08%, and 87.49% of the data have a standard deviation within 5, 15, and 25 ppm, respectively for the data collected at 1s. When aggregated to 1 min, the data showed 66.67%, 91.66%, and 94.44% of the data have a within 5, 15, and 25 ppm of each other, respectively.

This indicates that slightly less than half of the 1-s data measured by the sensors are within 5 ppm of each other  and that we can expect a majority of the data (>88.85%) to have errors up to 15 ppm depending on where on the car the inlet is mounted. When examining the  variability of the observed values for the 1 min data,  86.3% and 98.63% of the data have an  standard deviation within 5 and 25 ppm. In summary, the sampling location is a source of much greater uncertainty than instrument accuracy, drift, or linearity in the context of this work.

[revised manuscript text omitted]